# Visual Autoregressive Modeling: Scalable Image Generation via Next-Scale Prediction

**Keyu Tian**[1,2], **Yi Jiang**[2,†], **Zehuan Yuan**[2,*], **Bingyue Peng**[2], **Liwei Wang**[1,3,*]

[1]Center for Data Science, Peking University    [2]Bytedance Inc.
[3]State Key Lab of General Artificial Intelligence, School of
Intelligence Science and Technology, Peking University

keyutian@stu.pku.edu.cn, jiangyi.enjoy@bytedance.com,
yuanzehuan@bytedance.com, bingyue.peng@bytedance.com, wanglw@pku.edu.cn

Try and explore our online demo at: https://var.vision
Codes and models: https://github.com/FoundationVision/VAR

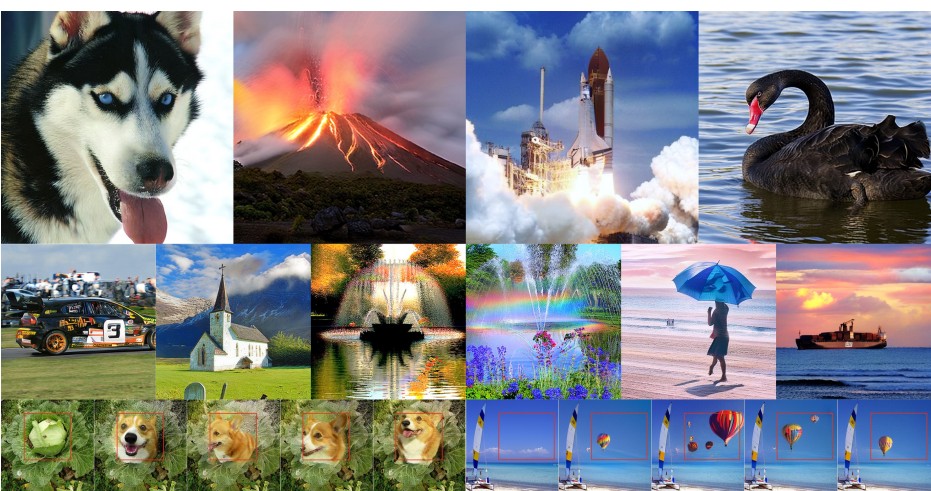

Figure 1: **Generated samples from Visual AutoRegressive (VAR) transformers trained on ImageNet**. We show 512×512 samples (top), 256×256 samples (middle), and zero-shot image editing results (bottom).

## Abstract

We present Visual AutoRegressive modeling (VAR), a new generation paradigm that redefines the autoregressive learning on images as coarse-to-fine "next-scale prediction" or "next-resolution prediction", diverging from the standard raster-scan "next-token prediction". This simple, intuitive methodology allows autoregressive (AR) transformers to learn visual distributions fast and can generalize well: VAR, for the *first time*, makes GPT-style AR models surpass diffusion transformers in image generation. On ImageNet 256×256 benchmark, VAR significantly improve AR baseline by improving Fréchet inception distance (FID) from 18.65 to 1.73, inception score (IS) from 80.4 to 350.2, with 20× faster inference speed. It is also empirically verified that VAR outperforms the Diffusion Transformer (DiT) in multiple dimensions including image quality, inference speed, data efficiency, and scalability. Scaling up VAR models exhibits clear power-law scaling laws similar to those observed in LLMs, with linear correlation coefficients near −0.998 as solid evidence. VAR further showcases zero-shot generalization ability in downstream tasks including image in-painting, out-painting, and editing. These results suggest VAR has initially emulated the two important properties of LLMs: **Scaling Laws** and **zero-shot** generalization. We have released all models and codes to promote the exploration of AR/VAR models for visual generation and unified learning.

---

*corresponding authors: wanglw@pku.edu.cn, yuanzehuan@bytedance.com; †: project lead

38th Conference on Neural Information Processing Systems (NeurIPS 2024).

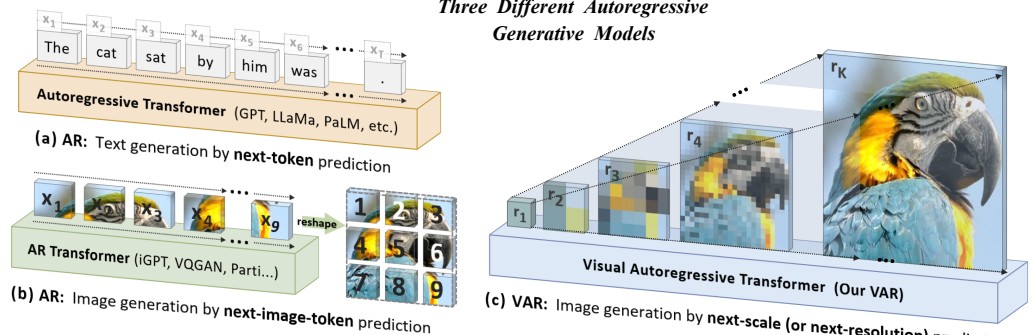

Figure 2: **Standard autoregressive modeling (AR) *vs.* our proposed visual autoregressive modeling (VAR).** (a) AR applied to language: sequential text token generation from left to right, word by word; (b) AR applied to images: sequential visual token generation in a raster-scan order, from left to right, top to bottom; (c) VAR for images: multi-scale token maps are autoregressively generated from coarse to fine scales (lower to higher resolutions), with parallel token generation within each scale. VAR requires a multi-scale VQVAE to work.

## 1 Introduction

The advent of GPT series [66, 67, 15, 63, 1] and more autoregressive (AR) large language models (LLMs) [22, 4, 39, 83, 84, 91, 79, 5, 80] has heralded a new epoch in the field of artificial intelligence. These models exhibit promising intelligence in generality and versatility that, despite issues like hallucinations [40], are still considered to take a solid step toward the general artificial intelligence (AGI). At the core of these models is a self-supervised learning strategy – *predicting the next token* in a sequence, a simple yet profound approach. Studies into **the success of these large AR models** have highlighted their **scalability and generalizabilty**: the former, as exemplified by *scaling laws* [44, 36], allows us to predict large model's performance from smaller ones and thus guides better resource allocation, while the latter, as evidenced by *zero-shot and few-shot* learning [67, 15], underscores the unsupervised-trained models' adaptability to diverse, unseen tasks. These properties reveal AR models' potential in learning from vast unlabeled data, encapsulating the essence of "AGI".

In parallel, the field of computer vision has been striving to develop large autoregressive or world models [59, 58, 6], aiming to emulate their impressive scalability and generalizability. Trailblazing efforts like VQGAN and DALL-E [30, 68] along with their successors [69, 92, 51, 99] have showcased the potential of AR models in image generation. These models utilize a visual tokenizer to discretize continuous images into grids of 2D tokens, which are then flattened to a 1D sequence for AR learning (Fig. 2 b), mirroring the process of sequential language modeling (Fig. 2 a). However, the scaling laws of these models remain underexplored, and more frustratingly, their performance **significantly lags** behind diffusion models [64, 3, 52], as shown in Fig. 3. In contrast to the remarkable achievements of LLMs, the power of autoregressive models in computer vision appears to be somewhat **locked**.

Autoregressive modeling requires defining the order of data. Our work reconsiders how to "order" an image: Humans typically perceive or create images in a hierachical manner, first capturing the global structure and then local details. This **multi-scale, coarse-to-fine** nature suggests an "order" for images. Also inspired by the widespread multi-scale designs [55, 53, 82, 45], we define autoregressive learning for images as "next-scale prediction" in Fig. 2 (c), diverging from the conventional "next-token prediction" in Fig. 2 (b). Our approach begins by encoding an image into multi-scale token maps. The autoregressive process is then started from the $1 \times 1$ token map, and progressively expands in resolution: at each step, the transformer predicts the next higher-resolution token map conditioned on all previous ones. We refer to this methodology as Visual AutoRegressive (VAR) modeling.

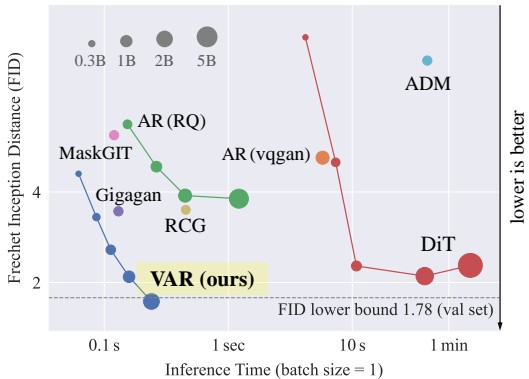

Figure 3: **Scaling behavior** of different model families on ImageNet $256 \times 256$ generation benchmark. The FID of the validation set serves as a reference lower bound (1.78). VAR with 2B parameters reaches an FID of 1.73, surpassing L-DiT with 3B or 7B parameters.

VAR directly leverages GPT-2-like transformer architecture [67] for visual autoregressive learning. On the ImageNet 256×256 benchmark, VAR significantly improves its AR baseline, achieving a Fréchet inception distance (FID) of 1.73 and an inception score (IS) of 350.2, with inference speed 20× faster (see Sec. 6 for details). Notably, VAR surpasses the Diffusion Transformer (DiT) – the foundation of leading diffusion systems like Stable Diffusion 3.0 and SORA [29, 14] – in FID/IS, data efficiency, inference speed, and scalability. VAR models also exhibit scaling laws akin to those witnessed in LLMs. Lastly, we showcase VAR's zero-shot generalization capabilities in tasks like image in-painting, out-painting, and editing. In summary, our contributions to the community include:

1. A new visual generative framework using a multi-scale autoregressive paradigm with next-scale prediction, offering new insights in autoregressive algorithm design for computer vision.

2. An empirical validation of VAR models' Scaling Laws and zero-shot generalization potential, which initially emulates the appealing properties of large language models (LLMs).

3. A breakthrough in visual autoregressive model performance, making GPT-style autoregressive methods surpass strong diffusion models in image synthesis *for the first time*[2].

4. A comprehensive open-source code suite, including both VQ tokenizer and autoregressive model training pipelines, to help propel the advancement of visual autoregressive learning.

## 2 Related Work

### 2.1 Properties of large autoregressive language models

**Scaling laws** are found and studied in *autoregressive* language models [44, 36], which describe a power-law relationship between the scale of model (or dataset, computation, *etc.*) and the cross-entropy loss value on the test set. Scaling laws allow us to directly predict the performance of a larger model from smaller ones [1], thus guiding better resource allocation. More pleasingly, they show that the performance of LLMs can scale well with the growth of model, data, and computation and never saturate, which is considered a key factor in the success of [15, 83, 84, 98, 91, 39]. The success brought by scaling laws has inspired the vision community to explore more similar methods for multimodality understanding and generation [54, 2, 89, 27, 96, 78, 21, 23, 42, 32, 33, 81, 88].

**Zero-shot generalization.** Zero-shot generalization [73] refers to the ability of a model, particularly a Large Language Model, to perform tasks that it has not been explicitly trained on. Within the realm of the computer vision, there is a burgeoning interest in the zero-shot and in-context learning abilities of foundation models, CLIP [65], SAM [49], Dinov2 [62]. Innovations like Painter [90] and LVM [6] extend visual prompters [41, 11] to achieve in-context learning in vision.

### 2.2 Visual generation

**Raster-scan autoregressive models** for visual generation necessitate the encoding of 2D images into 1D token sequences. Early endeavors [20, 85] have shown the ability to generate RGB (or grouped) pixels in the standard row-by-row, raster-scan manner. [70] extends [85] by using multiple independent trainable networks to do super-resolution repeatedly. VQGAN [30] advances [20, 85] by doing autoregressive learning in the latent space of VQVAE [86]. It employs GPT-2 decoder-only transformer to generate tokens in the raster-scan order, like how ViT [28] serializes 2D images into 1D patches. VQVAE-2 [69] and RQ-Transformer [51] also follow this raster-scan manner but use extra scales or stacked codes. Parti [93], based on the architecture of ViT-VQGAN [92], scales the transformer to 20B parameters and works well in text-to-image synthesis.

**Masked-prediction model.** MaskGIT [17] employs a VQ autoencoder and a masked prediction transformer similar to BERT [25, 10, 35] to generate VQ tokens through a greedy algorithm. MagViT [94] adapts this approach to videos, and MagViT-2 [95] enhances [17, 94] by introducing an improved VQVAE for both images and videos. MUSE [16] further scales MaskGIT to 3B parameters.

**Diffusion models**' progress has centered around improved learning or sampling [77, 76, 56, 57, 7], guidance [38, 61], latent learning [71], and architectures [37, 64, 72, 31]. DiT and U-ViT [64, 8] replaces or integrates the U-Net with transformer, and inspires recent image [19, 18] or video synthesis systems [12, 34] including Stable Diffusion 3.0 [29], SORA [14], and Vidu [9].

---

[2]A related work [95] named "language model beats diffusion" belongs to BERT-style masked-prediction model.

## 3 Method

### 3.1 Preliminary: autoregressive modeling via next-token prediction

**Formulation.** Consider a sequence of discrete tokens $x = (x_1, x_2, \ldots, x_T)$, where $x_t \in [V]$ is an integer from a vocabulary of size $V$. The next-token autoregressive posits the probability of observing the current token $x_t$ depends only on its prefix $(x_1, x_2, \ldots, x_{t-1})$. This **unidirectional token dependency assumption** allows for the factorization of the sequence $x$'s likelihood:

$$p(x_1, x_2, \ldots, x_T) = \prod_{t=1}^{T} p(x_t \mid x_1, x_2, \ldots, x_{t-1}). \tag{1}$$

Training an autoregressive model $p_\theta$ involves optimizing $p_\theta(x_t \mid x_1, x_2, \ldots, x_{t-1})$ over a dataset. This is known as the "next-token prediction", and the trained $p_\theta$ can generate new sequences.

**Tokenization.** Images are inherently 2D continuous signals. To apply autoregressive modeling to images via next-token prediction, we must: 1) tokenize an image into several *discrete* tokens, and 2) define a 1D *order* of tokens for unidirectional modeling. For 1), a quantized autoencoder such as [30] is often used to convert the image feature map $f \in \mathbb{R}^{h \times w \times C}$ to discrete tokens $q \in [V]^{h \times w}$:

$$f = \mathcal{E}(im), \qquad q = \mathcal{Q}(f), \tag{2}$$

where $im$ denotes the raw image, $\mathcal{E}(\cdot)$ a encoder, and $\mathcal{Q}(\cdot)$ a quantizer. The quantizer typically includes a learnable codebook $Z \in \mathbb{R}^{V \times C}$ containing $V$ vectors. The quantization process $q = \mathcal{Q}(f)$ will map each feature vector $f^{(i,j)}$ to the code index $q^{(i,j)}$ of its nearest code in the Euclidean sense:

$$q^{(i,j)} = \left( \arg\min_{v \in [V]} \|\text{lookup}(Z, v) - f^{(i,j)}\|_2 \right) \in [V], \tag{3}$$

where $\text{lookup}(Z, v)$ means taking the $v$-th vector in codebook $Z$. To train the quantized autoencoder, $Z$ is looked up by every $q^{(i,j)}$ to get $\hat{f}$, the approximation of original $f$. Then a new image $\hat{im}$ is reconstructed using the decoder $\mathcal{D}(\cdot)$ given $\hat{f}$, and a compound loss $\mathcal{L}$ is minimized:

$$\hat{f} = \text{lookup}(Z, q), \qquad \hat{im} = \mathcal{D}(\hat{f}), \tag{4}$$

$$\mathcal{L} = \|im - \hat{im}\|_2 + \|f - \hat{f}\|_2 + \lambda_P \mathcal{L}_P(\hat{im}) + \lambda_G \mathcal{L}_G(\hat{im}), \tag{5}$$

where $\mathcal{L}_P(\cdot)$ is a perceptual loss such as LPIPS [97], $\mathcal{L}_G(\cdot)$ a discriminative loss like StyleGAN's discriminator loss [47], and $\lambda_P$, $\lambda_G$ are loss weights. Once the autoencoder $\{\mathcal{E}, \mathcal{Q}, \mathcal{D}\}$ is fully trained, it will be used to tokenize images for subsequent training of a unidirectional autoregressive model.

The image tokens in $q \in [V]^{h \times w}$ are arranged in a 2D grid. Unlike natural language sentences with an inherent left-to-right ordering, the order of image tokens must be explicitly defined for unidirectional autoregressive learning. Previous AR methods [30, 92, 51] flatten the 2D grid of $q$ into a 1D sequence $x = (x_1, \ldots, x_{h \times w})$ using some strategy such as row-major raster scan, spiral, or z-curve order. Once flattened, they can extract a set of sequences $x$ from the dataset, and then train an autoregressive model to maximize the likelihood in (1) via next-token prediction.

**Discussion on the weakness of vanilla autoregressive models.** The above approach of tokenizing and flattening enable next-token autoregressive learning on images, but introduces several issues:

1) **Mathematical premise violation.** In quantized autoencoders (VQVAEs), the encoder typically produces an image feature map $f$ with inter-dependent feature vectors $f^{(i,j)}$ for all $i, j$. So after quantization and flattening, the token sequence $(x_1, x_2, \ldots, x_{h \times w})$ retains *bidirectional* correlations. This contradicts the *unidirectional* dependency assumption of autoregressive models, which dictates that each token $x_t$ should only depend on its prefix $(x_1, x_2, \ldots, x_{t-1})$.

2) **Inability to perform some zero-shot generalization.** Similar to issue 1), The unidirectional nature of image autoregressive modeling restricts their generalizability in tasks requiring bidirectional reasoning. E.g., it cannot predict the top part of an image given the bottom part.

3) **Structural degradation.** The flattening disrupts the spatial locality inherent in image feature maps. For example, the token $q^{(i,j)}$ and its 4 immediate neighbors $q^{(i\pm1,j)}$, $q^{(i,j\pm1)}$ are closely correlated due to their proximity. This spatial relationship is compromised in the linear sequence $x$, where *uni*directional constraints diminish these *cor*relations.

4) **Inefficiency.** Generating an image token sequence $x = (x_1, x_2, \ldots, x_{n \times n})$ with a conventional self-attention transformer incurs $\mathcal{O}(n^2)$ autoregressive steps and $\mathcal{O}(n^6)$ computational cost.

Issues 2) and 3) are evident (see examples above). Regarding issue 1), we present empirical evidence in Appendix C. The proof of issue 3) is detailed in Appendix D. These theoretical and practical limitations call for a rethinking of autoregressive models in the context of image generation.

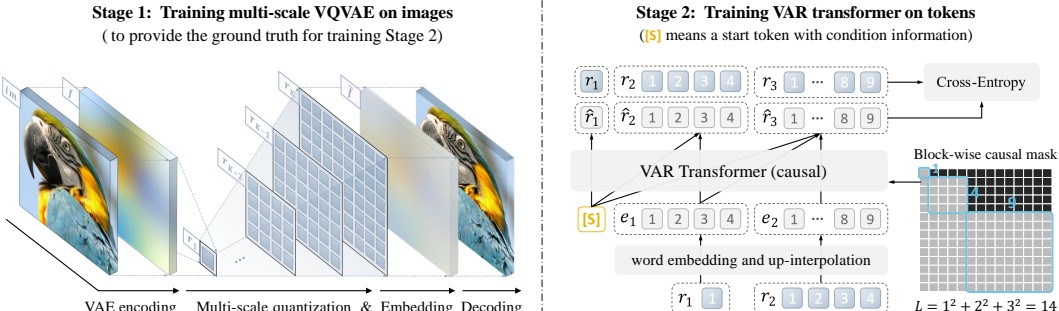

Figure 4: **VAR involves two separated training stages. Stage 1:** a multi-scale VQ autoencoder encodes an image into $K$ token maps $R = (r_1, r_2, \ldots, r_K)$ and is trained by a compound loss (5). For details on "Multi-scale quantization" and "Embedding", check Algorithm 1 and 2. **Stage 2:** a VAR transformer is trained via next-scale prediction (6): it takes ([s], $r_1, r_2, \ldots, r_{K-1}$) as input to predict $(r_1, r_2, r_3, \ldots, r_K)$. The attention mask is used in training to ensure each $r_k$ can only attend to $r_{\leq k}$. Standard cross-entropy loss is used.

## 3.2 Visual autoregressive modeling via next-scale prediction

**Reformulation.** We reconceptualize the autoregressive modeling on images by shifting from "next-token prediction" to "next-scale prediction" strategy. Here, the autoregressive unit is *an entire token map*, rather than *a single token*. We start by quantizing a feature map $f \in \mathbb{R}^{h \times w \times C}$ into $K$ multi-scale token maps $(r_1, r_2, \ldots, r_K)$, each at a increasingly higher resolution $h_k \times w_k$, culminating in $r_K$ matches the original feature map's resolution $h \times w$. The autoregressive likelihood is formulated as:

$$p(r_1, r_2, \ldots, r_K) = \prod_{k=1}^{K} p(r_k \mid r_1, r_2, \ldots, r_{k-1}), \tag{6}$$

where each autoregressive unit $r_k \in [V]^{h_k \times w_k}$ is the token map at scale $k$ containing $h_k \times w_k$ tokens, and the sequence $(r_1, r_2, \ldots, r_{k-1})$ serves as the the "prefix" for $r_k$. During the $k$-th autoregressive step, all distributions over the $h_k \times w_k$ tokens in $r_k$ will be generated in parallel, conditioned on $r_k$'s prefix and associated $k$-th position embedding map. This "next-scale prediction" methodology is what we define as visual autoregressive modeling (VAR), depicted on the right side of Fig. 4. Note that in the training of VAR, a block-wise causal attention mask is used to ensure that each $r_k$ can only attend to its prefix $r_{\leq k}$. During inference, kv-caching can be used and no mask is needed.

**Discussion.** VAR addresses the previously mentioned three issues as follows:

1) The mathematical premise is satisfied if we constrain each $r_k$ to depend only on its prefix, that is, the process of getting $r_k$ is solely related to $r_{\leq k}$. This constraint is acceptable as it aligns with the natural, coarse-to-fine progression characteristics like human visual perception and artistic drawing (as we discussed in Sec. 1). Further details are provided in the *Tokenization* below.

2) The spatial locality is preserved as (i) there is no flattening operation in VAR, and (ii) tokens in each $r_k$ are fully correlated. The multi-scale design additionally reinforces the spatial structure.

3) The complexity for generating an image with $n \times n$ latent is significantly reduced to $\mathcal{O}(n^4)$, see Appendix for proof. This efficiency gain arises from the *parallel* token generation in each $r_k$.

**Tokenization.** We develop a new multi-scale quantization autoencoder to encode an image to $K$ multi-scale discrete token maps $R = (r_1, r_2, \ldots, r_K)$ necessary for VAR learning (6). We employ the same architecture as VQGAN [30] but with a modified multi-scale quantization layer. The encoding and decoding procedures with residual design on $f$ or $\hat{f}$ are detailed in algorithms 1 and 2. We empirically find this residual-style design, akin to [51], can perform better than independent interpolation. Algorithm 1 shows that each $r_k$ would depend only on its prefix $(r_1, r_2, \ldots, r_{k-1})$.

Note that a shared codebook $Z$ is utilized across all scales, ensuring that each $r_k$'s tokens belong to the same vocabulary $[V]$. To address the information loss in upscaling $z_k$ to $h_K \times w_K$, we use $K$ extra convolution layers $\{\phi_k\}_{k=1}^K$. No convolution is used after downsampling $f$ to $h_k \times w_k$.

---

**Algorithm 1:** Multi-scale VQVAE Encoding

1 **Inputs:** raw image $im$;
2 **Hyperparameters:** steps $K$, resolutions $(h_k, w_k)_{k=1}^K$;
3 $f = \mathcal{E}(im), R = []$;
4 **for** $k = 1, \cdots, K$ **do**
5     $r_k = \mathcal{Q}(\text{interpolate}(f, h_k, w_k))$;
6     $R = \text{queue\_push}(R, r_k)$;
7     $z_k = \text{lookup}(Z, r_k)$;
8     $z_k = \text{interpolate}(z_k, h_K, w_K)$;
9     $f = f - \phi_k(z_k)$;
10 **Return:** multi-scale tokens $R$;

---

**Algorithm 2:** Multi-scale VQVAE Reconstruction

1 **Inputs:** multi-scale token maps $R$;
2 **Hyperparameters:** steps $K$, resolutions $(h_k, w_k)_{k=1}^K$;
3 $\hat{f} = 0$;
4 **for** $k = 1, \cdots, K$ **do**
5     $r_k = \text{queue\_pop}(R)$;
6     $z_k = \text{lookup}(Z, r_k)$;
7     $z_k = \text{interpolate}(z_k, h_K, w_K)$;
8     $\hat{f} = \hat{f} + \phi_k(z_k)$;
9 $\hat{im} = \mathcal{D}(\hat{f})$;
10 **Return:** reconstructed image $\hat{im}$;

---

## 4 Implementation details

**VAR tokenizer.** As aforementioned, we use the vanilla VQVAE architecture [30] and a multi-scale quantization scheme with $K$ extra convolutions (0.03M extra parameters). We use a shared codebook for all scales with $V = 4096$. Following the baseline [30], our tokenizer is also trained on OpenImages [50] with the compound loss (5) and a spatial downsample ratio of $16\times$.

**VAR transformer.** Our main focus is on VAR algorithm so we keep a simple model architecture design. We adopt the architecture of standard decoder-only transformers akin to GPT-2 and VQ-GAN [67, 30] with adaptive normalization (AdaLN), which has widespread adoption and proven effectiveness in many visual generative models [47, 48, 46, 75, 74, 43, 64, 19]. For class-conditional synthesis, we use the class embedding as the start token [s] and also the condition of AdaLN. We found normalizing $queries$ and $keys$ to unit vectors before attention can stablize the training. We do not use advanced techniques in large language models, such as rotary position embedding (RoPE), SwiGLU MLP, or RMS Norm [83, 84]. Our model shape follows a simple rule like [44] that the width $w$, head counts $h$, and drop rate $dr$ are linearly scaled with the depth $d$ as follows:

$$w = 64d, \qquad h = d, \qquad dr = 0.1 \cdot d/24. \tag{7}$$

Consequently, the main parameter count $N$ of a VAR transformer with depth $d$ is given by[3]:

$$N(d) = \underbrace{d \cdot 4w^2}_{\text{self-attention}} + \underbrace{d \cdot 8w^2}_{\text{feed-forward}} + \underbrace{d \cdot 6w^2}_{\text{adaptive layernorm}} = 18\,dw^2 = 73728\,d^3. \tag{8}$$

All models are trained with the similar settings: a base learning rate of $10^{-4}$ per 256 batch size, an AdamW optimizer with $\beta_1 = 0.9$, $\beta_2 = 0.95$, decay $= 0.05$, a batch size from 768 to 1024 and training epochs from 200 to 350 (depends on model size). The evaluations in Sec. 5 suggest that such a simple model design are capable of scaling and generalizing well.

## 5 Empirical Results

This section first compares VAR with other image generative model families in Sec. 5.1. Evaluations on the scalability and generalizability of VAR models are presented in Sec. 5.2 and Appendix B. For implementation details and ablation study, please see Appendix 4 and Appendix 6.

### 5.1 State-of-the-art image generation

**Setup.** We test VAR models with depths 16, 20, 24, and 30 on ImageNet $256\times256$ and $512\times512$ conditional generation benchmarks and compare them with the state-of-the-art image generation model families. Among all VQVAE-based AR or VAR models, VQGAN [30] and ours use the same architecture (CNN) and training data (OpenImages [50]) for VQVAE, while ViT-VQGAN [92] uses a ViT autoencoder, and both it and RQTransformer [51] trains the VQVAE directly on ImageNet. The results are summaried in Tab. 1 and Tab. 2.

---

[3]Due to resource limitation, we use a single shared adaptive layernorm (AdaLN) acorss all attention blocks in $512\times512$ synthesis. In this case, the parameter count would be reduced to around $12dw^2 + 6w^2 \approx 49152\,d^3$.

Table 1: **Generative model family comparison on class-conditional ImageNet 256×256**. "↓" or "↑" indicate lower or higher values are better. Metrics include Fréchet inception distance (FID), inception score (IS), precision (Pre) and recall (rec). "#Step": the number of model runs needed to generate an image. Wall-clock inference time relative to VAR is reported. Models with the suffix "-re" used rejection sampling. †: taken from MaskGIT [17].

| Type | Model | FID↓ | IS↑ | Pre↑ | Rec↑ | #Para | #Step | Time |
|------|-------|------|-----|------|------|-------|-------|------|
| GAN | BigGAN [13] | 6.95 | 224.5 | **0.89** | 0.38 | 112M | 1 | – |
| GAN | GigaGAN [43] | 3.45 | 225.5 | 0.84 | **0.61** | 569M | 1 | – |
| GAN | StyleGan-XL [75] | 2.30 | 265.1 | 0.78 | 0.53 | 166M | 1 | 0.3 [75] |
| Diff. | ADM [26] | 10.94 | 101.0 | 0.69 | 0.63 | 554M | 250 | 168 [75] |
| Diff. | CDM [37] | 4.88 | 158.7 | – | – | – | 8100 | – |
| Diff. | LDM-4-G [71] | 3.60 | 247.7 | – | – | 400M | 250 | – |
| Diff. | DiT-L/2 [64] | 5.02 | 167.2 | 0.75 | 0.57 | 458M | 250 | 31 |
| Diff. | DiT-XL/2 [64] | 2.27 | 278.2 | 0.83 | 0.57 | 675M | 250 | 45 |
| Diff. | L-DiT-3B [3] | 2.10 | 304.4 | 0.82 | 0.60 | 3.0B | 250 | >45 |
| Diff. | L-DiT-7B [3] | 2.28 | 316.2 | 0.83 | 0.58 | 7.0B | 250 | >45 |
| Mask. | MaskGIT [17] | 6.18 | 182.1 | 0.80 | 0.51 | 227M | 8 | 0.5 [17] |
| Mask. | RCG (cond.) [52] | 3.49 | 215.5 | – | – | 502M | 20 | 1.9 [52] |
| AR | VQVAE-2† [69] | 31.11 | ∼45 | 0.36 | 0.57 | 13.5B | 5120 | – |
| AR | VQGAN† [30] | 18.65 | 80.4 | 0.78 | 0.26 | 227M | 256 | 19 [17] |
| AR | VQGAN [30] | 15.78 | 74.3 | – | – | 1.4B | 256 | 24 |
| AR | VQGAN-re [30] | 5.20 | 280.3 | – | – | 1.4B | 256 | 24 |
| AR | ViTVQ [92] | 4.17 | 175.1 | – | – | 1.7B | 1024 | >24 |
| AR | ViTVQ-re [92] | 3.04 | 227.4 | – | – | 1.7B | 1024 | >24 |
| AR | RQTran. [51] | 7.55 | 134.0 | – | – | 3.8B | 68 | 21 |
| VAR | VAR-d16 | 3.30 | 274.4 | 0.84 | 0.51 | 310M | 10 | 0.4 |
| VAR | VAR-d20 | 2.57 | 302.6 | 0.83 | 0.56 | 600M | 10 | 0.5 |
| VAR | VAR-d24 | 2.09 | 312.9 | 0.82 | 0.59 | 1.0B | 10 | 0.6 |
| VAR | VAR-d30 | 1.92 | 323.1 | 0.82 | 0.59 | 2.0B | 10 | 1 |
| VAR | VAR-d30-re | **1.73** | **350.2** | 0.82 | 0.60 | 2.0B | 10 | 1 |
| | (validation data) | 1.78 | 236.9 | 0.75 | 0.67 | | | |

**Overall comparison.** In comparison with existing generative approaches including generative adversarial networks (GAN), diffusion models (Diff.), BERT-style masked-prediction models (Mask.), and GPT-style autoregressive models (AR), our visual autoregressive (VAR) establishes a new model class. As shown in Tab. 1, VAR not only achieves the best FID/IS but also demonstrates remarkable speed in image generation. VAR also maintains decent precision and recall, confirming its semantic consistency. These advantages hold true on the 512×512 synthesis benchmark, as detailed in Tab. 2. Notably, VAR significantly advances traditional AR capabilities. To our knowledge, this is the *first time* of autoregressive models outperforming Diffusion transformers, a milestone made possible by VAR's resolution of AR limitations discussed in Section 3.

**Efficiency comparison.** Conventional autoregressive (AR) models [30, 69, 92, 51] suffer a lot from the high computational cost, as the number of image tokens is quadratic to the image resolution. A full autoregressive generation of $n^2$ tokens requires $\mathcal{O}(n^2)$ decoding iterations and $\mathcal{O}(n^6)$ total computations. In contrast, VAR only requires $\mathcal{O}(\log(n))$ iterations and $\mathcal{O}(n^4)$ total computations. The wall-clock time reported in Tab. 1 also provides empirical evidence that VAR is around 20 times faster than VQGAN and ViT-VQGAN even with more model parameters, reaching the speed of efficient GAN models which only require 1 step to generate an image.

Table 2: **ImageNet 512×512 conditional generation.** †: quoted from MaskGIT [17]. "-s": a single shared AdaLN layer is used due to resource limitation.

| Type | Model | FID↓ | IS↑ | Time |
|------|-------|------|-----|------|
| GAN | BigGAN [13] | 8.43 | 177.9 | – |
| Diff. | ADM [26] | 23.24 | 101.0 | – |
| Diff. | DiT-XL/2 [64] | 3.04 | 240.8 | 81 |
| Mask. | MaskGIT [17] | 7.32 | 156.0 | 0.5† |
| AR | VQGAN [30] | 26.52 | 66.8 | 25† |
| VAR | VAR-d36-s | **2.63** | **303.2** | 1 |

**Compared with popular diffusion transformer.** The VAR model surpasses the recently popular diffusion models Diffusion Transformer (DiT), which serves as the precursor to the latest Stable-Diffusion 3 [29] and SORA [14], in multiple dimensions: 1) In image generation diversity and quality

(FID and IS), VAR with 2B parameters consistently performs better than DiT-XL/2 [64], L-DiT-3B, and L-DiT-7B [3]. VAR also maintains comparable precision and recall. 2) For inference speed, the DiT-XL/2 requires 45× the wall-clock time compared to VAR, while 3B and 7B models [3] would cost much more. 3) VAR is considered more data-efficient, as it requires only 350 training epochs compared to DiT-XL/2's 1400. 4) For scalability, Fig. 3 and Tab. 1 show that DiT only obtains marginal or even negative gains beyond 675M parameters. In contrast, the FID and IS of VAR are consistently improved, aligning with the scaling law study in Sec. 5.2. These results establish *VAR as potentially a more efficient and scalable model for image generation than models like DiT*.

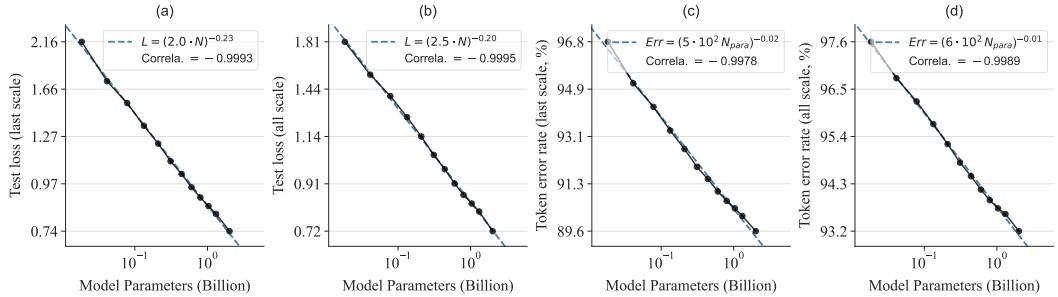

Figure 5: **Scaling laws with VAR transformer size** $N$, with power-law fits (dashed) and equations (in legend). Small, near-zero exponents $\alpha$ suggest a smooth decline in both test loss $L$ and token error rate $Err$ when scaling up VAR transformer. Axes are all on a logarithmic scale. The Pearson correlation coefficients near $-0.998$ signify a strong linear relationship between $log(N)$ *vs.* $log(L)$ or $log(N)$ *vs.* $log(Err)$.

## 5.2 Power-law scaling laws

**Background.** Prior research [44, 36, 39, 1] have established that scaling up autoregressive (AR) large language models (LLMs) leads to a predictable decrease in test loss $L$. This trend correlates with parameter counts $N$, training tokens $T$, and optimal training compute $C_{\min}$, following a power-law:

$$L = (\beta \cdot X)^{\alpha}, \tag{9}$$

where $X$ can be any of $N$, $T$, or $C_{\min}$. The exponent $\alpha$ reflects the smoothness of power-law, and $L$ denotes the reducible loss normalized by irreducible loss $L_{\infty}$ [36] A logarithmic transformation to $L$ and $X$ will reveal a linear relation between $\log(L)$ and $\log(X)$:

$$\log(L) = \alpha \log(X) + \alpha \log \beta. \tag{10}$$

An appealing phenomenon is that both [44] and [36] never observed deviation from these linear relationships at the higher end of $X$, although flattening is inevitable as the loss approaches zero.

These observed scaling laws [44, 36, 39, 1] not only validate the scalability of LLMs but also serve as a predictive tool for AR modeling, which facilitates the estimation of performance for larger AR models based on their smaller counterparts, thereby saving resource usage by large model performance forecasting. Given these appealing properties of scaling laws brought by LLMs, their replication in computer vision is therefore of significant interest.

**Setup of scaling VAR models.** Following the protocols from [44, 36, 39, 1], we examine whether our VAR model complies with similar scaling laws. We trained models across 12 different sizes, from 18M to 2B parameters, on the ImageNet training set [24] containing 1.28M images (or 870B image tokens under our VQVAE) per epoch. For models of different sizes, training spanned 200 to 350 epochs, with a maximum number of tokens reaching 305 billion. Below we focus on the scaling laws with model parameters $N$ and optimal training compute $C_{\min}$ given sufficient token count $T$.

**Scaling laws with model parameters** $N$**.** We first investigate the test loss trend as the VAR model size increases. The number of parameters $N(d) = 73728\, d^3$ for a VAR transformer with depth $d$ is specified in (8). We varied $d$ from 6 to 30, yielding 12 models with 18.5M to 2.0B parameters. We assessed the final test cross-entropy loss $L$ and token prediction error rates $Err$ on the ImageNet validation set of 50,000 images [24]. We computed $L$ and $Err$ for both the last scale (at the last next-scale autoregressive step), as well as the global average. Results are plotted in Fig. 5, where we

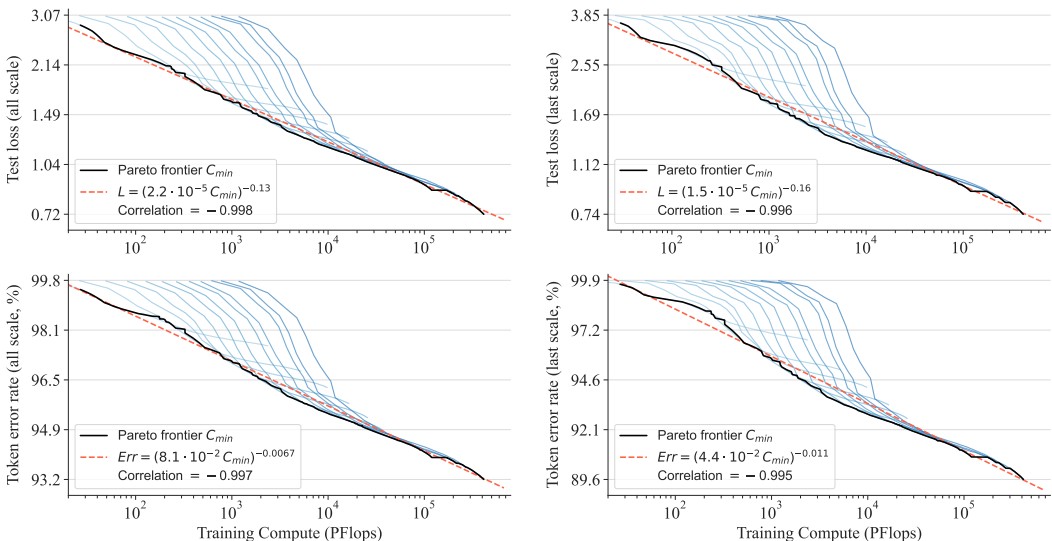

Figure 6: **Scaling laws with optimal training compute $C_{\min}$.** Line color denotes different model sizes. Red dashed lines are power-law fits with equations in legend. Axes are on a logarithmic scale. Pearson coefficients near $-0.99$ indicate strong linear relationships between $\log(C_{\min})$ *vs.* $\log(L)$ or $\log(C_{\min})$ *vs.* $\log(Err)$.

observed a clear power-law scaling trend for $L$ as a function of $N$, as consistent with [44, 36, 39, 1]. The power-law scaling laws can be expressed as:

$$L_{\text{last}} = (2.0 \cdot N)^{-0.23} \quad \text{and} \quad L_{\text{avg}} = (2.5 \cdot N)^{-0.20}. \tag{11}$$

Although the scaling laws are mainly studied on the test loss, we also empirically observed similar power-law trends for the token error rate $Err$:

$$Err_{\text{last}} = (4.9 \cdot 10^2 N)^{-0.016} \quad \text{and} \quad Err_{\text{avg}} = (6.5 \cdot 10^2 N)^{-0.010}. \tag{12}$$

These results verify the strong scalability of VAR, by which scaling up VAR transformers can continuously improve the model's test performance.

**Scaling laws with optimal training compute $C_{\min}$.** We then examine the scaling behavior of VAR transformers when increasing training compute $C$. For each of the 12 models, we traced the test loss $L$ and token error rate $Err$ as a function of $C$ during training quoted in PFlops ($10^{15}$ floating-point operations per second). The results are plotted in Fig. 6. Here, we draw the Pareto frontier of $L$ and $Err$ to highlight the optimal training compute $C_{\min}$ required to reach a certain value of loss or error.

The fitted power-law scaling laws for $L$ and $Err$ as a function of $C_{\min}$ are:

$$L_{\text{last}} = (2.2 \cdot 10^{-5} C_{\min})^{-0.13} \tag{13}$$

$$L_{\text{avg}} = (1.5 \cdot 10^{-5} C_{\min})^{-0.16}, \tag{14}$$

$$Err_{\text{last}} = (8.1 \cdot 10^{-2} C_{\min})^{-0.0067} \tag{15}$$

$$Err_{\text{avg}} = (4.4 \cdot 10^{-2} C_{\min})^{-0.011}. \tag{16}$$

These relations (14, 16) hold across 6 orders of magnitude in $C_{\min}$, and our findings are consistent with those in [44, 36]: when trained with sufficient data, larger VAR transformers are more compute-efficient because they can reach the same level of performance with less computation.

## 6 Ablation Study

In this study, we aim to verify the effectiveness and efficiency of our proposed VAR framework. Results are reported in Tab. 3.

**Effectiveness and efficiency of VAR.** Starting from the vanilla AR transformer baseline implemented by [17], we replace its methodology with our VAR and keep other settings unchanged to get row 2.

Table 3: **Ablation study of VAR.** The first two rows compare GPT-2-style transformers trained under AR or VAR algorithm without any bells and whistles. Subsequent lines show the influence of VAR enhancements. "AdaLN": adaptive layernorm. "CFG": classifier-free guidance. "Attn. Norm.": normalizing $q$ and $k$ to unit vectors before attention. "Cost": inference cost relative to the baseline. "$\Delta$": FID reduction to the baseline.

|   | Description | Para. | Model | AdaLN | Top-$k$ | CFG | Cost | FID↓ | $\Delta$ |
|---|---|---|---|---|---|---|---|---|---|
| 1 | AR [30] | 227M | AR | ✗ | ✗ | ✗ | 1 | 18.65 | 0.00 |
| 2 | AR to VAR | 207M | VAR-$d$16 | ✗ | ✗ | ✗ | 0.013 | 5.22 | $-13.43$ |
| 3 | +AdaLN | 310M | VAR-$d$16 | ✓ | ✗ | ✗ | 0.016 | 4.95 | $-13.70$ |
| 4 | +Top-$k$ | 310M | VAR-$d$16 | ✓ | 900 | ✗ | 0.016 | 4.64 | $-14.01$ |
| 5 | +CFG | 310M | VAR-$d$16 | ✓ | 900 | 1.5 | 0.022 | 3.60 | $-15.05$ |
| 5 | +Attn. Norm. | 310M | VAR-$d$16 | ✓ | 900 | 1.5 | 0.022 | 3.30 | $-15.35$ |
| 6 | +Scale up | 2.0B | VAR-$d$30 | ✓ | 900 | 1.5 | 0.052 | 1.73 | $-16.85$ |

VAR achieves a way more better FID (18.65 *vs.* 5.22) with only $0.013\times$ inference wall-clock cost than the AR model, which demonstrates a leap in visual AR model's performance and efficiency.

**Component-wise ablation.** We further test some key components in VAR. By replacing the standard Layer Normalization (LN) with Adaptive Layer Normalization (AdaLN), VAR starts yielding better FID than baseline. By using the top-$k$ sampling similar to the baseline, VAR's FID is further improved. By using the classifier-free guidance (CFG) with ratio 1.5 and normalizing $q$ and $k$ to unit vectors before attention, we reach the FID of 3.30, which is 15.35 lower to the baseline, and its inference speed is still 45 times faster. We finally scale up VAR size to 2.0B and achieve an FID of 1.73. This is 16.85 better than the baseline FID.

## 7 Limitations and Future Work

In this work, we mainly focus on the design of learning paradigm and keep the VQVAE architecture and training unchanged from the baseline [30] to better justify VAR framework's effectiveness. We expect **advancing VQVAE tokenizer** [99, 60, 95] as another promising way to enhance autoregressive generative models, which is orthogonal to our work. We believe iterating VAR by advanced tokenizer or sampling techniques in these latest work can further improve VAR's performance or speed.

**Text-prompt generation** is an ongoing direction of our research. Given that our model is fundamentally similar to modern LLMs, it can easily be integrated with them to perform text-to-image generation through either an encoder-decoder or in-context manner.

**Video generation** is not implemented in this work, but it can be naturally extended. By considering multi-scale video features as **3D pyramids**, we can formulate a similar "**3D next-scale prediction**" to generate videos via VAR. Compared to diffusion-based generators like SORA [14], our method has inherent advantages in temporal consistency or integration with LLMs, thus can potentially handle longer temporal dependencies. This makes VAR competitive in the video generation field, because traditional AR models can be too inefficient for video generation due to their extremely high computational complexity and slow inference speed: it is becoming prohibitively expensive to generate high-resolution videos with traditional AR models, while VAR is capable to solve this. We therefore foresee a promising future for exploiting VAR models in the realm of video generation.

## 8 Conclusion

We introduced a new visual generative framework named Visual AutoRegressive modeling (VAR) that 1) theoretically addresses some issues inherent in standard image autoregressive (AR) models, and 2) makes language-model-based AR models first surpass strong diffusion models in terms of image quality, diversity, data efficiency, and inference speed. Upon scaling VAR to 2 billion parameters, we observed a clear power-law relationship between test performance and model parameters or training compute, with Pearson coefficients nearing $-0.998$, indicating a robust framework for performance prediction. These scaling laws and the possibility for zero-shot task generalization, as hallmarks of LLMs, have now been initially verified in our VAR transformer models. We hope our findings and open sources can facilitate a more seamless integration of the substantial successes from the natural language processing domain into computer vision, ultimately contributing to the advancement of powerful multi-modal intelligence.

# 9 Acknowledgements

Liwei Wang was supported by National Science Foundation of China (NSFC92470123, NSFC62276005) and National Science and Technology Major Project (2022ZD0114902).

# A Visualization of scaling effect

To better understand how VAR models are learning when scaled up, we compare some generated $256 \times 256$ samples from VAR models of 4 different sizes (depth 6, 16, 26, 30) and 3 different training stages (20%, 60%, 100% of total training tokens) in Fig. 7. To keep the content consistent, a same random seed and teacher-forced initial tokens are used. The observed improvements in visual fidelity and soundness are consistent with the scaling laws, as larger transformers are thought able to learn more complex and fine-grained image distributions.

# B Zero-shot task generalization

**Image in-painting and out-painting.** VAR-$d30$ is tested. For in- and out-painting, we teacher-force ground truth tokens outside the mask and let the model only generate tokens within the mask. No class label information is injected into the model. The results are visualized in Fig. 8. Without modifications to the network architecture or tuning parameters, VAR has achieved decent results on these downstream tasks, substantiating the generalization ability of VAR.

**Class-conditional image editing.** Following MaskGIT [17] we also tested VAR on the class-conditional image editing task. Similar to the case of in-painting, the model is forced to generate tokens only in the bounding box conditional on some class label. Fig. 8 shows the model can produce plausible content that fuses well into the surrounding contexts, again verifying the generality of VAR.

# C Token dependency in VQVAE

To examine the token dependency in VQVAE [30], we check the attention scores in the self-attention layer before the vector quantization module. We randomly sample 4 $256 \times 256$ images from the ImageNet validation set for this analysis. Note the self-attention layer in [30] only has 1 head so for each image we just plot one attention map. The heat map in Fig. 9 shows the attention scores of each token to all other tokens, which indicate a strong, bidirectional dependency among all tokens. This is not surprising since the VQVAE model, trained to reconstruct images, leverages self-attention layers without any attention mask. Some work [87] has used causal attention in self-attention layers of a video VAE, but we did not find any image VAE work uses causal self-attention.

# D Time complexity of AR and VAR generation

We prove the time complexity of AR and VAR generation.

**Lemma D.1.** *For a standard self-attention transformer, the time complexity of AR generation is $\mathcal{O}(n^6)$, where $h = w = n$ and $h, w$ are the height and width of the VQ code map, respectively.*

*Proof.* The total number of tokens is $h \times w = n^2$. For the $i$-th ($1 \leq i \leq n^2$) autoregressive iteration, the attention scores between each token and all other tokens need to be computed, which requires $\mathcal{O}(i^2)$ time. So the total time complexity would be:

$$\sum_{i=1}^{n^2} i^2 = \frac{1}{6}n^2(n^2 + 1)(2n^2 + 1), \tag{17}$$

Which is equivalent to $\mathcal{O}(n^6)$ basic computation. $\square$

For VAR, it needs us to define the resolution sequence $(h_1, w_1, h_2, w_2, \ldots, h_K, w_K)$ for autoregressive generation, where $h_i, w_i$ are the height and width of the VQ code map at the $i$-th autoregressive step, and $h_K = h, w_K = w$ reaches the final resolution. Suppose $n_k = h_k = w_k$ for all $1 \leq k \leq K$

Scaling up training compute $C$

Scaling up transformer parameters $N$

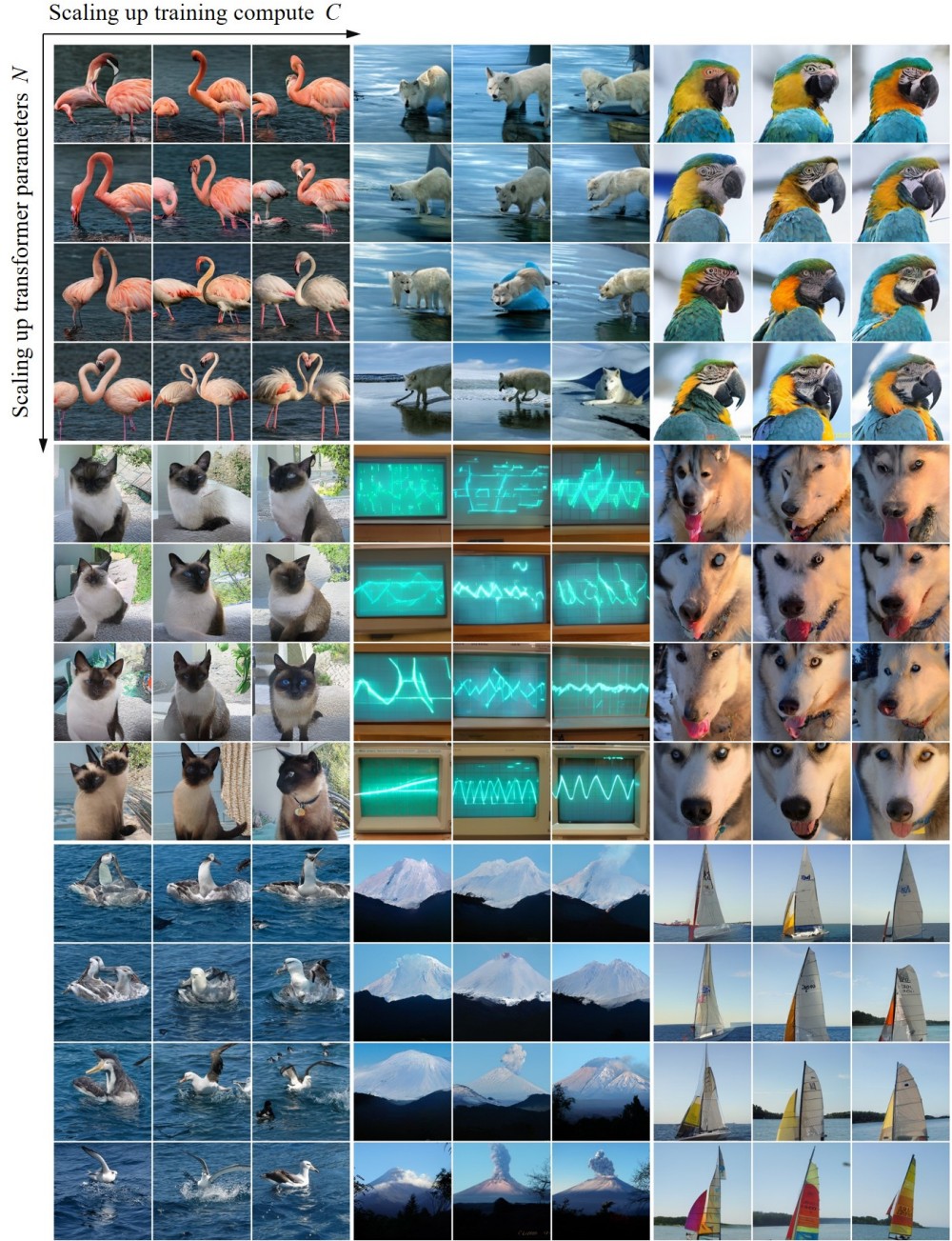

Figure 7: **Scaling model size $N$ and training compute $C$ improves visual fidelity and soundness.** Zoom in for a better view. Samples are drawn from VAR models of 4 different sizes and 3 different training stages. 9 class labels (from left to right, top to bottom) are: flamingo 130, arctic wolf 270, macaw 88, Siamese cat 284, oscilloscope 688, husky 250, mollymawk 146, volcano 980, and catamaran 484.

and $n = h = w$, for simplicity. We set the resolutions as $n_k = a^{(k-1)}$ where $a > 1$ is a constant such that $a^{(K-1)} = n$.

**Lemma D.2.** *For a standard self-attention transformer and given hyperparameter $a > 1$, the time complexity of VAR generation is $\mathcal{O}(n^4)$, where $h = w = n$ and $h, w$ are the height and width of the last (largest) VQ code map, respectively.*

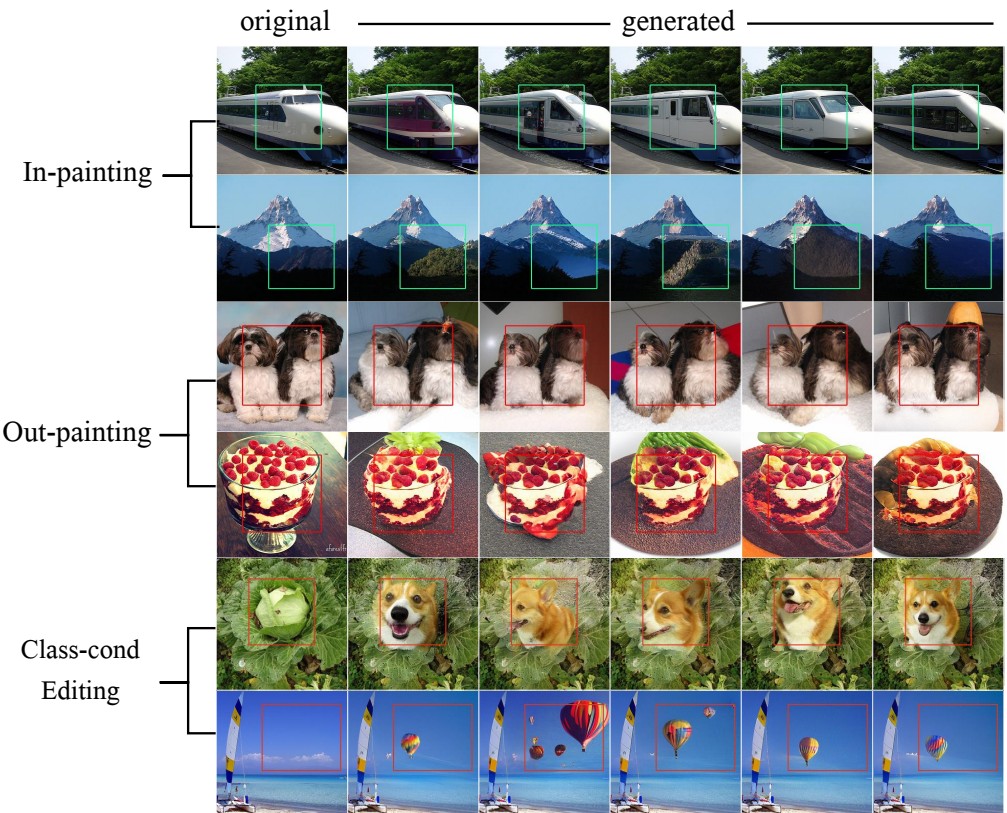

Figure 8: **Zero-shot evaluation in downstream tasks** containing in-painting, out-painting, and class-conditional editing. The results show that VAR can generalize to novel downstream tasks without special design and finetuning. Zoom in for a better view.

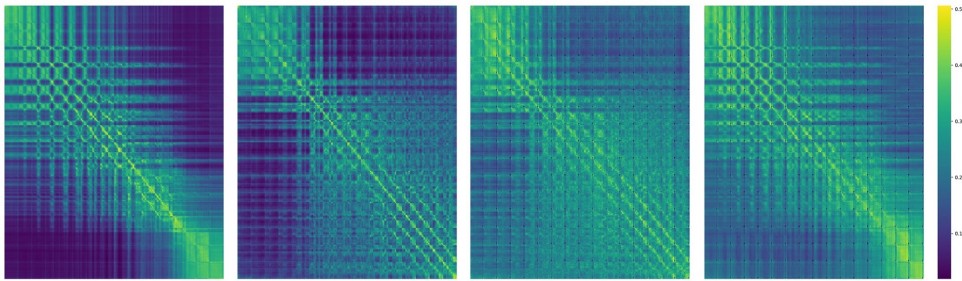

Figure 9: **Token dependency plotted.** The normalized heat map of attention scores in the last self-attention layer of VQGAN encoder is visualized. 4 random 256×256 images from ImageNet validation set are used.

*Proof.* Consider the $k$-th ($1 \leq k \leq K$) autoregressive generation. The total number of tokens of current all token maps $(r_1, r_2, \ldots, r_k)$ is:

$$\sum_{i=1}^{k} n_i^2 = \sum_{i=1}^{k} a^{2 \cdot (k-1)} = \frac{a^{2k} - 1}{a^2 - 1}. \tag{18}$$

So the time complexity of the $k$-th autoregressive generation would be:

$$\left( \frac{a^{2k} - 1}{a^2 - 1} \right)^2. \tag{19}$$

By summing up all autoregressive generations, we have:

$$\sum_{k=1}^{\log_a(n)+1} \left(\frac{a^{2k}-1}{a^2-1}\right)^2 \tag{20}$$

$$= \frac{(a^4-1)\log n + \left(a^8 n^4 - 2a^6 n^2 - 2a^4(n^2-1) + 2a^2 - 1\right)\log a}{(a^2-1)^3(a^2+1)\log a} \tag{21}$$

$$\sim \mathcal{O}(n^4). \tag{22}$$

This completes the proof. $\qquad\square$

BigGAN (FID=6.95)    VQVAE-2 (FID=31)    MaskGIT (FID=6.18)    VAR, ours (FID=1.92)

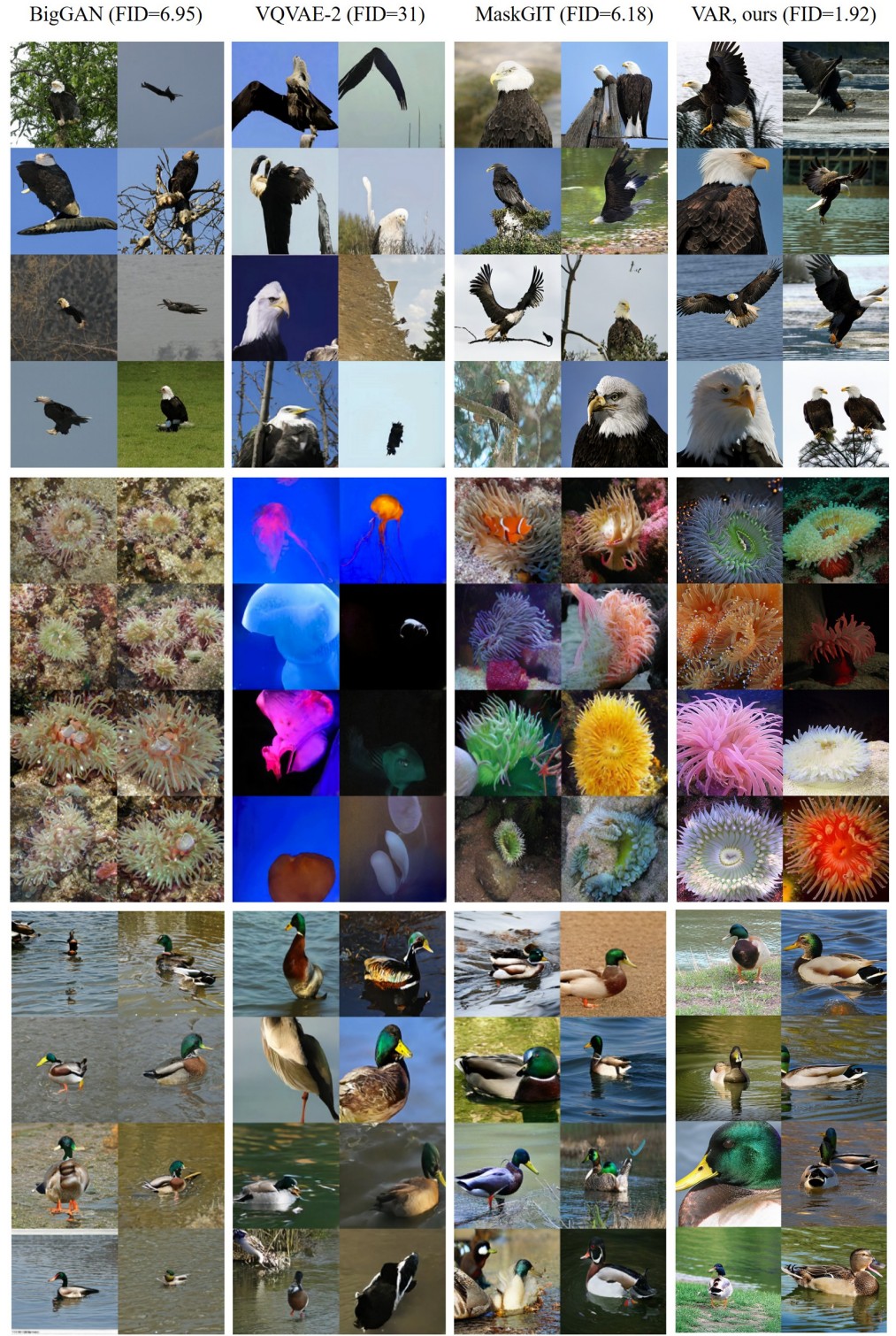

Figure 10: **Model comparison on ImageNet 256×256 benchmark.** More generated 512×512 samples by VAR can be found in the submitted Supplementary Material zip file.

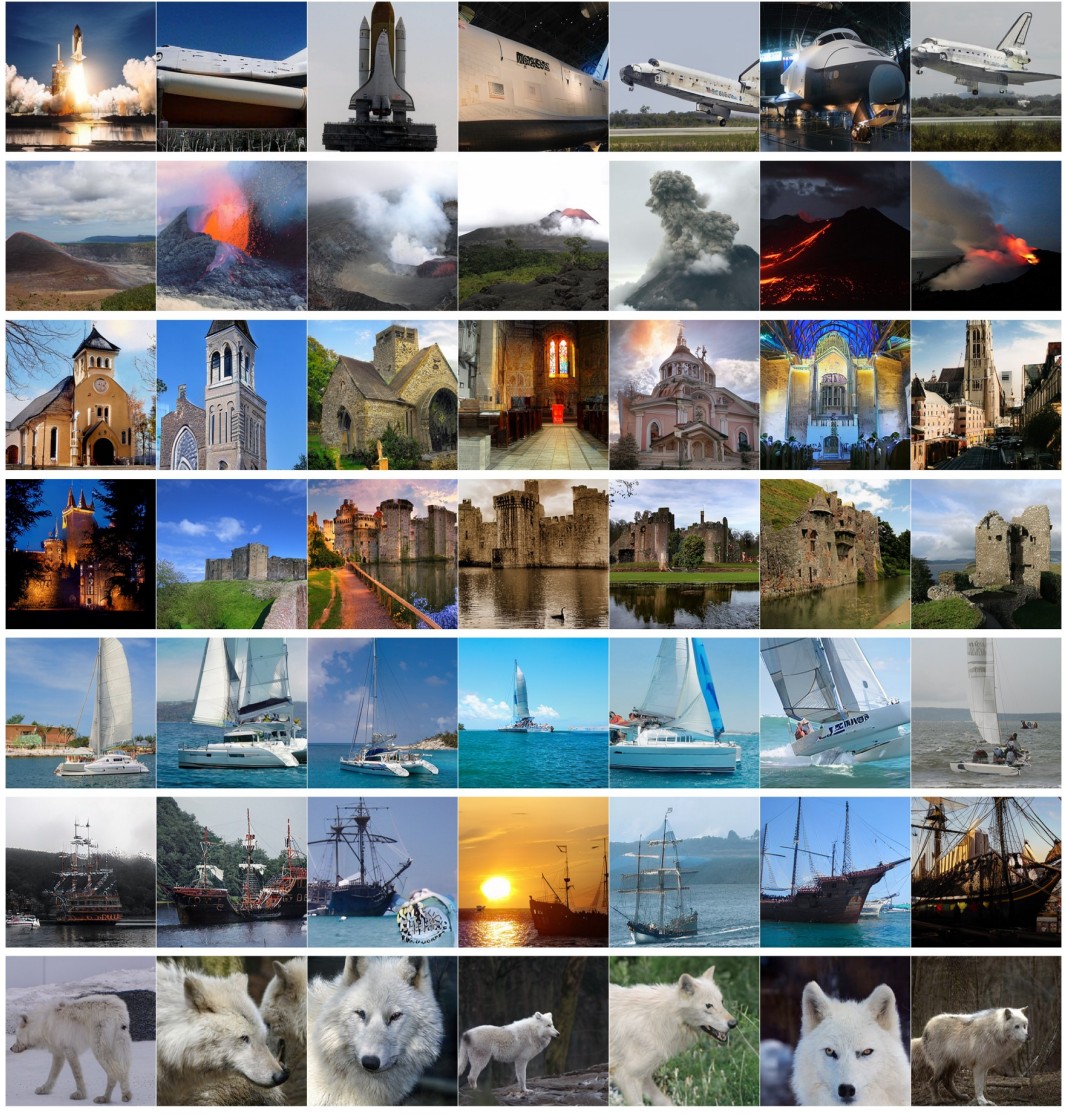

Figure 11: **Some generated 256×256 samples by VAR trained on ImageNet.** More generated 512×512 samples by VAR can be found in the submitted Supplementary Material zip file.

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
