# OpenReview forum: "Visual Autoregressive Modeling: Scalable Image Generation via Next-Scale Prediction"
_NeurIPS.cc/2024/Conference — NeurIPS 2024 oral_

### Official Review · Reviewer_dFai · 2024-07-03

**Soundness:** 4
**Presentation:** 3
**Contribution:** 4
**Rating:** 8
**Confidence:** 4

**Summary:**

The paper introduces VAR, a novel autoregressive generative model for images that treats each scale in a multi-resolution feature pyramid as a token. Unlike traditional models that predict the next token from a rasterized grid, VAR predicts the next scale in a multi-resolution grid. This approach demonstrates greater scalability than next-token prediction and extends the well-known scaling laws from language modeling to image generation. Extensive experiments show that VAR outperforms both diffusion and AR baselines while offering improved efficiency in both training and inference.

**Strengths:**

- The paper addresses the significant question of bridging the performance gap between autoregressive language models and autoregressive image generation. This makes the topic highly relevant for the community and potentially impactful.

- The method is well-motivated and utilizes well-known building blocks from LLMs. Hence, VAR is an important step toward showing that with a proper scheme, widely used LLM architectures can perform competitively in the Image domain.

- The experimental section is comprehensive, demonstrating VAR's performance and efficiency in image generation on ImageNet 256 and 512. Additionally, the authors provide in-depth discussion of scaling laws for VAR.

- Ablation studies in Appendix D clearly illustrate the contribution of different aspects of VAR to the final model.

**Weaknesses:**

- While the writing of the paper is clear for the most part, the method section could benefit from better presentation. The authors provide some details on VAR tokenization and training, but I found section 3, especially section 3.2, slightly confusing. I suggest that the authors add more details and clarification on VAR's training process, the residual tokenization, and the workings of the transformer part. Currently, these details are somewhat obscured in Algorithm 1 and 2, and Figure 4.

- Some claims in the paper are slightly exaggerated. For example, in the abstract, the authors mention that VAR brings the FID from 18.65 to 1.73. While this is true, the FID of 18.65 belongs to a relatively weak baseline for AR models. It would be better to rewrite such claims in relation to more realistic baselines, such as RQ-VAE models, which are closer to VAR's methodology

- The baselines used for the diffusion part are also relatively weak. For instance, MDTv2 [1] is a transformer-based diffusion model that achieves an FID of 1.58 on ImageNet 256. Therefore, it would be more appropriate to state that VAR performs "competitively" with diffusion models rather than significantly outperforms them.

[1] Gao S, Zhou P, Cheng MM, Yan S. MDTv2: Masked Diffusion Transformer is a Strong Image Synthesizer. arXiv preprint arXiv:2303.14389. 2023 Mar 25.

**Questions:**

1. Does VAR perform next-scale prediction completely in the latent space? For example, if the image resolution is 256 and VQ-VAE has a latent size of 32, does VAR operate on resolutions up to 32, or does it extend up to the image resolution?

2. Is there a loss term missing in equation (5)? VQ-GAN models typically include a commitment loss and a codebook loss, but it appears VAR's quantizer only uses one of these losses. Could you provide more details on this part?

3. If I understand correctly, when predicting the probabilities for each scale with a transformer, the transformer now needs to estimate a much higher-dimensional distribution (a distribution over the entire grid instead of just one point in the grid). Can you provide some intuition on why the transformer part can handle this prediction task effectively? Is it due to the strong conditioning from the lower scales?

4. Are the features computed by the VAR quantizer also useful for image recognition tasks, or do they perform best in image generation?

**Limitations:**

The authors have addressed limitations and social impact of the work.

---

> ### Author Rebuttal · Authors · 2024-08-07
>
> Dear Reviewer dFai,
>
> Many thanks to your professional, detailed, and valuable reviews. We're going to response to your concerns one by one.
>
> -----------------
>
> > [W1] I suggest that the authors add more details and clarification on VAR's training process, the residual tokenization, and the workings of the transformer part. Currently, these details are somewhat obscured in Algorithm 1 and 2, and Figure 4.
>
> [A1] Thank you for your thorough review and suggestion. We've rearranged our Method part, adding more details on the two different training stages, and make them more closed to each other.
>
> &nbsp;
>
> > [W2] It would be better to rewrite such claims in relation to more realistic baselines, such as RQ-VAE models, which are closer to VAR's methodology.
>
> [A2] We agree that comparing RQ-VAE with VAR can also demonstrates the effectiveness and efficiency of VAR and without any potential overclaims. Also, this would not devalue VAR's novelty and technical contributions. We have updated these according descriptions.
>
> &nbsp;
>
> > [W3] Therefore, it would be more appropriate to state that VAR performs "competitively" with diffusion models rather than significantly outperforms them.
>
> [A3] Thanks for this professional advice. We'd first explain why models like MDTv2 was not inclued in  table 1: Since our table 1 mainly focuses on **Generative model family** comparison on class-conditional ImageNet 256x256, we did not include some latest powerful models in it. As also mentioned in line 308, our main focus was on VAR algorithm and we used a plain GPT-2 transformer without SwiGLU, RMSNorm, or Rotary position embedding. So there is still large room to boost VAR. Comparing VAR with other long-optimized diffusion model can be a bit unfair. We have added some explanations and used "performs competitively" to describe our VAR.
>
> Nonetheless, we added an extra comparison which focuses on comparing VAR with latest, state-of-the-art model to the Appendix. We also updated the Future Work section to see if we can integrate more advanced technique like in MDTv2 or LLaMa to further upgrade VAR.
>
> &nbsp;
>
> -------------------------------
>
> **Thanks for your comments and suggestions, we will add these experiments to our revision. Feel free to let us know if you have any further questions or concerns :-).**

---

> ### Author Response · Authors · 2024-08-07
> **Response to Questions 1-4.**
>
> > [Q1] Does VAR perform next-scale prediction completely in the latent space? For example, if the image resolution is 256 and VQ-VAE has a latent size of 32, does VAR operate on resolutions up to 32, or does it extend up to the image resolution?
>
> [A4] Yes, VAR is completely done in the latent space. As mentioned in line307, our VQVAE uses a downsampling ratio of 16. So the largest latet token map $r_K$ of 256px|512px image would have a size of 16|32.
>
> &nbsp;
>
>
> > [Q2] Is there a loss term missing in equation (5)? VQ-GAN models typically include a commitment loss and a codebook loss, but it appears VAR's quantizer only uses one of these losses. Could you provide more details on this part?
>
> [A5] The equation (5) is a simplified version of the loss function in VQ-GAN. We use the same VQVAE training loss as VQGAN so we also have a commitment loss and a codebook loss too. We'll update equation (5) to clarify this.
>
> &nbsp;
>
> > [Q3] Can you provide some intuition on why the transformer part can handle this prediction task effectively? Is it due to the strong conditioning from the lower scales?
>
> [A6] Recall our motivation in line41: "Our work reconsiders how to order an image.
> Humans typically perceive or create images in a hierachical manner, first capturing the global structure and then local details.
> This \textbf{multi-scale, coarse-to-fine} nature suggests an order for images".
> If this is acknowledged, then predicting multiple tokens at the same time is natural: it just mimics how human understands or creates images.
>
> On the other hand, the model and computation scaling is also vital to VAR's good performance. As continuing to scale-up the size of transformer and improving the expressive power of model, the model will have stronger capability to solve difficult task like VAR pretraining.
>
> &nbsp;
>
> > [Q4] Are the features computed by the VAR quantizer also useful for image recognition tasks, or do they perform best in image generation?
>
> [A7] Thanks for such an interesting and enlightening question. Exploring whether VARs can improve image understanding similar to previous image pre-training work (e.g., contrastive learning and masked modelling) is a highly valuable direction. We have also included this in the Future Work section and will actively explore it in the future!

---

> ### Comment · Reviewer_dFai · 2024-08-07
> **Response to the rebuttal**
>
> I would like to thank the authors for taking the time to answer my comments and for the clarifications. I believe that VAR is a strong work that should be highlighted at the conference. Hence, I would like to keep my initial score.

---

> > ### Author Response · Authors · 2024-08-08
> > **Thanks for your timely reply and the kind words!**
> >
> > Thank you for your quickly reply! We will remain active until the discussion period ends. Please feel free to get back to us if you have any new questions :-)!

---

### Official Review · Reviewer_yx9H · 2024-07-09

**Soundness:** 3
**Presentation:** 4
**Contribution:** 4
**Rating:** 8
**Confidence:** 5

**Summary:**

The paper introduces Visual AutoRegressive modeling (VAR), which uses a coarse-to-fine approach for image generation. VAR drastically improves performance, reducing FID from 18.65 to 1.73 and increasing IS from 80.4 to 350.2, with 20x faster inference. It outperforms diffusion transformers in quality, speed, efficiency, and scalability. VAR models show scaling laws like large language models and demonstrate zero-shot generalization in image editing tasks.

**Strengths:**

- **Solid Motivation**: The scale in vision signals is a natural choice for autoregressive generation. The exploration of autoregression in visual generation is indeed a worthy topic.
- **Novel Method**: This work is the first to explore a visual generative framework using a multi-scale autoregressive paradigm with next-scale prediction.
- **Strong Performance**: The paper demonstrates significant advancements in visual autoregressive model performance, with GPT-style autoregressive methods surpassing strong diffusion models in image synthesis for the first time.
- **Promising Scaling Law**: The paper presents a promising scaling law for the proposed visual autoregressive modeling paradigm.

**Weaknesses:**

- **Lack of Ablation Study on VQVAE**: There is no ablation study on the newly proposed VQ-VAE model. In Table 3, the performance differences between the first two rows cannot be solely attributed to the model change from AR to VAR, as the VQ-VAE model has also been modified.
- **Resolution Flexibility**: The resolutions for VAR generation appear to be pre-defined and bound to the VQ-VAE model during its pre-training. Adjusting the number of resolutions or maximum resolution without re-training the VQ-VAE model seems non-trivial.

**Questions:**

**About VQVAE**

- What is the rFID of your VQVAE? Can you provide a table comparing your VQVAE and VQVAEs of other works (e.g., VQGAN, MaskGIT)

- What is the pre-training cost of your VQVAE?

- If residual quantization is not used, and instead a multi-scale token map is constructed directly by some ways like: 1) Independently downsample the VAE encoder features to multiple scales, then quantize each scale directly, or 2) Construct multiple resolution ImageNet datasets (e.g., ImageNet 16x16, 32x32…, 256x256) and independently apply vector quantization to each dataset, thereby obtaining low-resolution to high-resolution token maps.Then, can the proposed visual autoregressive modeling still be performed? In other words, is residual quantization an *indispensable* part of VAR modeling? (Ignoring the efficiency or complexity of these alternative tokenization methods)
- What is the impact on performance if residual quantization is not used?

**Question on Figure 7**

The apples-to-apples qualitative comparison, like in Figure 7, is common in diffusion-based models because the initial noise is of the same resolution as the final output, allowing significant control over the final image when coupled with deterministic sampling. However, in VAR, the counterpart to the "initial noise" is only the "teacher-forced initial tokens," which, if I understand correctly, are of 1x1 size. This suggests only a very loose control over the final image. Given this, why doesn't this result in a situation similar to the "butterfly effect," where identical initial states and random seeds, due to different model configurations, lead to significantly different final outputs after multiple generation iterations?

**Question on zero-shot generation**

- Is only the last resolution of the token map masked and then teacher-forced to generate the masked regions, or is each token map masked?
- The model hasn't explicitly learned inter-token dependence at the spatial level during training. Could you provide an explanation of why the model can perform zero-shot editing/inpainting, which requires the model to condition on tokens at some spatial positions to generate tokens at other positions?

**Limitations:**

This paper validates the effectiveness of VAR only in class-conditional generation scenarios. Applying VAR to text-to-image generation is a worthwhile area for future exploration.

---

> ### Author Rebuttal · Authors · 2024-08-07
>
> Dear Reviewer yx9H,
>
> ---------------------------
>
> > [W1 Q1, about VQVAE]
>
> [A1] We appreciate your detailed comments and will address them one by one.
>
> 1. VQVAE rFID: please see the overall Author Rebuttal part of "VQVAE concerns"
> 2. pre-training cost: please see the overall Author Rebuttal part of "VQVAE concerns"
> 3. why we didn't choose other multi-scale quantization ways: if the residual quantization is removed and **independently quantization** is used (e.t., independently downsampling VAE features, or independently encoding image in 16x16, 32x32, ..., 256x256),
> the mathematical premise of unidirectional dependency would be broken. The details can be found in line140 of our paper.
> To ensure that property, we have to use the **residual quantization** way. We'll add these explanations to our manuscript.
> 4. Lack of Ablation Study on VQVAE: please see the overall Author Rebuttal part of "VQVAE concerns".
>
> &nbsp;
>
> > [W2] Resolution Flexibility
>
> [A2] Thanks for this detailed comment. We've observed that the multi-scale VQVAE pretrained only on 256px images, can easily generalize to higher resolutions like 512 and 1024. We have added visualizations of this to a new Appendix part.
>
> > [Q2] Question on Figure 7
>
> [A3] Thank you for this professional question.
> In practice, we: 1) use a fixed sampling seed for every step, and 2) also fixed the first token which intuitively determined the global structure of the image.
> We found both of them are crucial for the generation consistency.
> We'll add these details to our manuscript.
>
> &nbsp;
>
> > [Q3] Question on zero-shot generation
>
> [A4] For the details on our zero-shot generalisation algorithm, please see the overall Author Rebuttal part of "Zero-shot generalisation algorithm".
> For your second question, we actually allows full, bidirectional attention among tokens in the same scale $r_k$.
> So the model can learn inter-token dependence.
>
> &nbsp;
>
> -----------------------------------
>
> **Lastly, thank you so much for helping us improve the paper and appreciate your open discussions! Please let us know if you have any further questions. We are actively available until the end of this rebuttal period. Looking forward to hearing back from you!**

---

> ### Comment · Reviewer_yx9H · 2024-08-11
>
> ### Regarding [A1]
>
> 1. What I'm concerned is *exactly* this premise. From my perspective, coarse-to-fine scaled images may indeed not strictly have causal dependency if not constructed in your way. However, is this **strict** causal dependency fundamental to the success of your method? Or is the core idea that predicting the coarse-scale image first, followed by the fine-scale image, the key to your success (even if the strict causal dependency doesn't hold, as in the example I provided)? I believe this is a critical factor that warrants an ablation study.
>
> 2. It appears that there is no ablation study on the "residual quantization" technique. Specifically, I am referring to the case where you still construct a multi-scale token map in your VQ-tokenizer, but without applying residual quantization.
>
> ### Regarding [A2]
>
> Could you provide more details on how the 256x256 tokenizer can be directly applied to tokenize a 512x512 image?
>
> ### Regarding [A4]
>
> I understand that bidirectional attention is allowed during training. However, the model’s objective is to use this bidirectional attention to learn how to *predict the next scale*, rather than *predicting some spatial positions based on others*. My point is that the second objective was not explicitly optimized during training—can you confirm if this is accurate?

---

> ### Author Response · Authors · 2024-08-11
> **Response to Reviewer yx9H**
>
> ### Regarding [A1]
>
> Yes, we found the causal dependency (which is brought by the residual quantization) is critical to VAR's success, just as the left-to-right linguistic order is important to LLM's success. Earlier in the VAR research, we had tried the two independent encoding ways you mentioned and compared them to the causal one:
>
> - Independently encoding images of different resolutions was nearly impossible because a 16x downsampled VQVAE could barely reconstruct an image with <=64 resolution. Huge color differences and shape deformation.
> - Independent quantization can be seen as the **ablation** on the "residual quantization". We had tried this idea and while the reconstruction quality of the VQVAE did not change much, the VAR's validation token accuracy decreased a lot (tested on an ImageNet subset with 250 classes):
>
>     |Method|Model|#Para|Accuracy$\uparrow$|
>     |:-|:-|:-|:-|
>     |Residual quantization|VAR-$d24$|1.0B|3.92%|
>     |Independently quantization|VAR-$d24$|1.0B|3.01%|
>
> - Empirically, a 0.9% accuracy drop implies a huge performance degradation when the model size was close to 1B (can be sensed in Figure 5(c)). Therefore, we abandoned this independent scheme early in our research.
>
> We do agree with you that "coarse-to-fine scaled images may indeed not strictly have causal dependency". **If they don't**, the whole generation process is more like a series of super-resolutions; **but if they do**, it's much more similar to the way human paintings work: first the whole, then the details, with each step being a **refinement** to all the past steps (due to the **residual**).
>
> We highly believe the **refinement** is a key to VAR's success since it allows VAR to fix past mistakes like a diffusion model. Among AR methods, this is an unique advantage of VAR because it is even impossible for language AR models to do this -- they can't correct historical mistakes via some "residual" mechanism. We'll add these extra ablation studies and discussions to our paper and thanks for your insightful questions.
>
> &nbsp;
>
> ### Regarding [A2]
>
> The tokenizer consists of three parts, the CNN encoder, the quantizer, and the CNN decoder. Multi-scale exists only in the quantizer, so we just need to change the resolution hyperparameters $h_1, h_2, \dots, h_K$ in the multiscale quantizer, detailed in Algorithm 1 and 2. Since the operations in it are interpolations and convolutions which can generalize to any resolution, no additional operations are needed anymore. In other words, we only need to set $h_1, h_2, \dots, h_K$ from (1,2,3,4,5,6,8,10,13,16) to (1,2,3,4,6,9,13,18,24,32) and Algorithm 1 and 2 will still work. We'll update our paper to make this more clear.
>
> &nbsp;
>
> ### Regarding [A4]
>
> Yes, we agree that "predicting some spatial positions based on others" is not direct learned through VAR (which may be learned through BERT, MAE, etc.). But we checked the VAR's self-attention score when it generated an image, and observed that many tokens on a certain object's body would still show relatively high attention scores. Taking this into account, and the fact that VAR can fully utilize information from all previous scales, it still has the ability to do tasks like inpainting (Figure 8). We'll add these explanations to our paper for better presentation.
>
> -----------------------------------
>
> Thank you again for your insightful and professional comment, which made our work more complete and solid! If there're any further questions, please let us know. If you feel all questions have been addressed, you can kindly consider re-rating our work. Thank you!

---

> > ### Comment · Reviewer_yx9H · 2024-08-11
> >
> > Thanks for your detailed reply.
> >
> > I have one additional question: diffusion-based models are able to flexibly change inference steps for a more detailed generation result (large steps) or a sketchy yet faster generation (small steps). Is VAR possible of achieving this? Or the inference step is somewhat constrained in VAR? Does this constitute a weakness of VAR?

---

> ### Author Response · Authors · 2024-08-11
> **Inference steps**
>
> Thanks for this question. Unlike VQGAN or other AR image generators, where the number of steps is usually fixed, the inference steps in VAR (also the scale shapes in its VQVAE) can be slightly adjusted, e.g., replacing (1,2,3,4,5,6,8,10,13,16) with (1,2,4,6,8,10,13,16) or (1,2,3,4,5,6,8,10,12,14,16). This is possible because VAR treats a scale, rather than a single token, as an autoregressive unit, and operations based on the unit (interpolations) can generalize to different spatial resolutions. In other words, the reason is that we chose the scales in the visual signal for autoregressive generation, which is also mentioned as the **Strengths1** and **Strengths2** in the review comment.
>
> Another natural question is how diffusion transformers will perform when the number of steps is small enough to approach VAR's 10 steps. We investigated this by using some ODE solvers like DDIM and DPM-Solver to reduce the diffusion inference steps. The results are as below.
>
> | Model                  | #Para | #Step | Time  | FID$\downarrow$ |
> |------------------------|-------|-------|-------|--------|
> | VAR-$d24$              | 1.0B  | 10    | **0.6**   | **2.09**   |
> | DiT-XL/2 (original)    | 675M  | 250   | 45    | 2.27   |
> | DiT-XL/2 + DDIM        | 675M  | 250   | 45    | 2.14   |
> | DiT-XL/2 + DDIM        | 675M  | 20    | 2.9   | 4.68   |
> | DiT-XL/2 + DDIM        | 675M  | 10    | 1.8   | 12.38  |
> | U-ViT-H/2 + DPM-solver | 501M  | 20    | 15.6  | 2.53   |
> | U-ViT-H/2 + DPM-solver | 501M  | 10    | 7.8   | 3.18   |
>
> It can be seen that while diffusion can be accelerated many times by the ODE solver, the sacrifice of FID becomes non-trivial when the number of steps reach 10.
>
> &nbsp;
>
> When considered more broadly, we feel there's a **common limitation** of all Diffusion, AR, and VAR models: it is not possible to **automatically** determine the number of steps based on the input. For example, generating a blank blackboard clearly requires a different number of steps than a blackboard filled with math formulas. AR/VAR are expected to solve this with some modifications, like introducing an [EOS] token to allow the model itself to predict when to stop. We believe this is a valuable direction to be explored and will add it into our Limitation or Future Work section.

---

> > ### Comment · Reviewer_yx9H · 2024-08-12
> >
> > The author has adequately addressed my concerns, and I am inclined to raise my score to 8. This is a commendable piece of work.

---

> > > ### Author Response · Authors · 2024-08-12
> > > **Thanks for your time and effort and kind words**
> > >
> > > We're glad to hear that your concerns have been adequately addressed. We appreciate your professional and constructive feedback which made our work more solid and clear.
> > >
> > > We'll revise the paper based on the discussions to better present VAR. If you have any questions, please feel free to lcomment here. Thank you.

---

### Official Review · Reviewer_oCJa · 2024-07-12

**Soundness:** 3
**Presentation:** 4
**Contribution:** 4
**Rating:** 8
**Confidence:** 4

**Summary:**

This work proposes a novel approach to image generation using an autoregressive decoder-only transformer model. Rather than decoding in a raster-scan their approach (VAR) decodes scales/resolutions conditioned on previously generated scales, reminiscent of traditional scale pyramids in computer vision. VAR demonstrates competitive performance on ImageNet in terms of generation quality, diversity and inference speed. It also demonstrates scaling laws up to 2.0B parameters.

**Strengths:**

- I am *very* impressed by the proposed method. It is simple, intuitive and novel. It is not difficult to see why it works well.
- The empirical results are strong, and since the approach is based on a decoder-only transformer (a tried and tested architecture for autoregressive generation) I would expect it to scale to foundation-model T2I systems easily.
- The paper is well written and easy to read, with the method/motivation/story communicated clearly to the reader.

**Weaknesses:**

As the reviewing burden has been heavy for this conference (6 papers) please understand that I can only dedicate so much time to this paper. Thus, I may have made mistakes in my understanding of the paper, and I welcome the authors to correct me if this is the case.

1. The reported inference efficiency is a bit disingenuous. The comparison with DiT uses 250 steps, which is much more than what SotA samplers require. Moreover, diffusion models can be distilled into 1-4 step models that are even faster.  Compared to raster scan autoregressive models, the improvement in latency seems to be due to better use of parallel resources. However, for larger batches/measuring throughput this advantage may fade.
2. There are some missing details and insight, especially with regards to the multi-scale VQVAE. There are also details that are present in the code that really should be in the paper, such as the number of tokens per image/scale. What is its reconstruction performance of the VQVAE (compared to e.g. the SDXL VAE)? How does one choose the number of scales?  How many codes end up being used over the different scales and do different scales capture similar (spatial) information in the latent space vs the image space? It would also be great to see an ablation like Table 3 for the VQVAE. Also, the code provided doesn't give details to reproduce the VQVAE, only the transformer.
3. The use of "zero-shot" to refer to the model's editing ability is different in nature to zero-shot generalisation in LLMs, so I find the link made in the paper to be a little disingenuous. Moreover, the editing performance doesn't seem to be very strong, with inpainting generation spilling outside of the box. The paper also doesn't give details on how the editing is performed.

**Questions:**

1. I'd like to see a throughput comparison in img/s as the batch size is increased. I'd also like to see some DiT results where a more advanced sampler such as DPM-solver or SA-solver is used.
2. See above.
3. Concretely, how is the editing performed? Are certain tokens teacher-forced? If so starting from which scale?

**Limitations:**

See above

---

> ### Author Rebuttal · Authors · 2024-08-07
>
> Dear Reviewer oCJa,
>
> Many thanks to your valuable comments and questions, which help us a lot to improve our work. We address your questions as follows.
>
> ---------------------------
>
> > [W1 Q1, about the efficiency evaluation] I'd like to see a throughput comparison in img/s as the batch size is increased. I'd also like to see some DiT results where a more advanced sampler such as DPM-solver or SA-solver is used.
>
> [A1] Thank you for your suggestions to make our efficiency evaluation more complete and thoughtful!
>
> First, we add the throughput comparison with batched inference to our manuscript.
> All models are tested on a single A100 GPU. VQGAN's results are quoted from [1].
> From the results we can see that VAR also **gets a big speedup from batch processing**, similarly to vanilla autoregressive models. We attribute this to the short sequence length of the first few VAR steps (like generating $1\times 1$ and $2\times 2$ tokens).
> After batching, VAR reaches a throughput of 0.04 $\text{img/s}$, which is still **4 times faster than batched VQGAN**, even if VAR has a larger model size.
>
> |Model|#Para|Batch size|Throughput$\downarrow$|FID$\downarrow$|
> |:-|:-|:-|:-|:-|
> | VQGAN-re     | 1.4B | 1   | 5.76 $\text{img/s}$  | 5.20 |
> | VQGAN-re     | 1.4B | 100 | 0.16 $\text{img/s}$  | 5.20 |
> | VAR-$d30$-re | 2.0B | 1   | 0.24 $\text{img/s}$  | 1.73 |
> | VAR-$d30$-re | 2.0B | 100 | **0.04 $\text{img/s}$**  | **1.73** |
> -----------------------
> Second, we add the results of DiT and U-ViT (another transformer-based diffusion model) with fewer diffusion steps ($<50$) to our manuscript.
> The results are quoted from [2, 3]. From the table one can see that:
> 1) Reducing the number of diffusion steps via ODE sampler (like DDIM, DPM-Solver) will result in a large FID rise, especially when reduced to near 10 steps.
> 2) Both using 10 steps, VAR-1B is still **faster and better than DiT-675M**. This is also because VAR has a shorter sequence length in the early steps, e.g., the first three VAR steps only generate $1\times 1 + 2\times 2 + 3\times 3 = 14$ tokens.
> 3) The Diffusion community has been building up for a long time for efficiency boosting. In contrast, VAR, as a newly proposed method, promises to see more ways to accelerate or distill VAR in the future. We'd like to add this to our Future Work section.
>
> |Model|#Para|#Step|Time$\downarrow$|FID$\downarrow$|
> |:-|:-|:-|:-|:-|
> | VAR-$d24$              | 1.0B | 10 | **0.6** | **2.09** |
> | DiT-XL/2 (original)    | 675M |250 | 45      | 2.27 |
> | DiT-XL/2 + DDIM        | 675M |250 | 45      | 2.14 |
> | DiT-XL/2 + DDIM        | 675M | 20 | 2.9     | 4.68 |
> | DiT-XL/2 + DDIM        | 675M | 10 | 1.8     | 12.38|
> | U-ViT-H/2 + DPM-solver | 501M | 20 | 15.6    | 2.53 |
> | U-ViT-H/2 + DPM-solver | 501M | 10 | 7.8     | 3.18 |
>
> > [W2 Q2, details on multi-scale VQVAE]
>
> [A2] We'd like to respond to your queries mentioned in Weakness 2 one by one, and add all the details below to Appendix section.
> Please see the overall Author Rebuttal part of "VQVAE concerns" for specific responses.
>
> &nbsp;
>
>
> > [W3 Q3, The use of "zero-shot" to refer to the model's editing ability is different in nature to zero-shot generalisation in LLMs; doesn't give details on how the editing is performed]
>
> [A3] Thanks for your insightful comments. In the field of language processing, it has been verified that every task can be formulated to an autoregressive generation task. This allows a pretrained LLM can generalize to many tasks different to its pretraining task without any finetuning.
> In our model, we also use the term "zero-shot" to emphasize that our model can also do tasks that are different to our pretraining task. We have updated some according descriptions in our paper to make it more clear and accurate.
>
> We added enough details on how the inpainting/outpainting/editing is performed to our manuscript. You can also find them in the overall Author Rebuttal part.
>
>
> -----------------------------------
>
> Many thanks to Reviewer oCJa for their professional, detailed, and valuable reviews!
> We have done our best to address each of your concerns and hope our response can resolve them.
> Please let us know if you have any other questions. We will actively join the discussion until the end of the rebuttal period.
> We are looking forward to hearing from you :-) !

---

> > ### Author Response · Authors · 2024-08-08
> > **References**
> >
> > References:
> >
> > [1] Chang H , Zhang H , Jiang L ,et al. MaskGIT: Masked Generative Image Transformer[J]. arXiv.2202.04200.
> >
> > [2] Bao F , Nie S , Xue K ,et al. All are Worth Words: A ViT Backbone for Diffusion Models[J]. arXiv.2209.12152.
> >
> > [3] Ma X, Fang G, Michael Bi,et al. Learning-to-Cache: Accelerating Diffusion Transformer via Layer Caching[J]. arXiv.2406.01733

---

> > ### Comment · Reviewer_oCJa · 2024-08-08
> >
> > Thank you for the rebuttal; I feel like most of my questions have been well addressed. The additional results with faster diffusion samplers have strengthened the paper.
> > I still have a couple of reservations:
> > - I'm a bit confused by the throughput results, there appears to be some sort of typo (?) as throughput should be better when higher and the VQGAN results are higher.
> > - If editing is performed using interpolation, how does one mask the boundary tokens that don't align with the mask at full resolution? E.g. in your example is the first token at the first scale teacher forced or not? My impression is that this lack of spatial resolution at lower scales probably limits fine-grained control over boundaries for the in/outpainting tasks. If this is indeed the case it would be best to mention this as a (current) limitation.

---

> ### Author Response · Authors · 2024-08-09
> **We're glad to know most questions were addressed; We reply to the two more questions**
>
> Thank you again for your professional review which did strengthen our paper. For the two more questions:
>
> - Yes it's a typo. All "img/s" should be "s/img". We've corrected this in our manuscript.
>
> - Yes, when the spatial resolution is too low, some interpolations could be inaccurate. But the smallest token map would be 2x2 because during inference the 1x1 was the start token. We've added this to our limitation.
>
> - Here we provide more details on that "interpolation": Considering an inpainting case on a 256x256 image where the upper left NxN pixels are masked. To mask the smallest 2x2 token map, the binary mask $M$ will be interpolated to 2x2 $M_2$. The upper left token on the 2x2 token map will be masked only if $M_2^{(0,0)} \ge 0.5$. We've added these to the Appendix to make it more clear.

---

> > ### Comment · Reviewer_oCJa · 2024-08-09
> >
> > Thanks for the clarification. Now it is clear to me how the editing is performed.
> >
> > I just noticed that the batch size is 100 < 256 which is the max number of tokens VAR infers in parallel for a single sample. Does the throughput improve for VQGAN if the batch size is increased further? To be clear I fully agree that VAR uses parallel compute much more efficiently than raster-scan autoregression for low batch sizes. However, my intuition tells me that in the case that VQGAN fully utilises the parallel compute of a GPU/accelerator then VAR should not have a throughput advantage (and VQGAN may even do better since it has fewer tokens to infer per sample).

---

> ### Author Response · Authors · 2024-08-09
> **Response to the batched inference**
>
> Thanks for the insightful question on the throughput improvement. We further investigate the batched inference using batch size of 150, 200, and 250:
>
> |Model|#Para|Batch size|Throughput$\downarrow$|FID$\downarrow$|
> |:-|:-|:-|:-|:-|
> | VQGAN-re     | 1.4B | 1   | 5.76 $\text{s/img}$  | 5.20 |
> | VQGAN-re     | 1.4B | 100 | 0.16 $\text{s/img}$  | 5.20 |
> | VQGAN-re     | 1.4B | 150 | 0.15 $\text{s/img}$  | 5.20 |
> | VQGAN-re     | 1.4B | 200 | 0.14 $\text{s/img}$  | 5.20 |
> | VQGAN-re     | 1.4B | 250 | OOM  | $-$ |
> | VAR-$d30$-re | 2.0B | 1   | 0.240 $\text{s/img}$  | 1.73 |
> | VAR-$d30$-re | 2.0B | 100 | 0.041 $\text{s/img}$  | 1.73 |
> | VAR-$d30$-re | 2.0B | 150 | **0.039 $\text{s/img}$**  | **1.73** |
> | VAR-$d30$-re | 2.0B | 200 | OOM  | $-$ |
>
> **Observation.** When the batch size gets larger than 100, the throughput improvements of both VQGAN and VAR becomes marginal. Upon reaching the largest batch size without out of memory issue, VAR still shows **3.6x throughput**, though it was larger and has more tokens to infer.
>
> **Analysis.** To understand why VQGAN does not present higher throughput than VAR when larger batch sizes are used, we further check the behaviors of VAR and VQGAN when they do autoregressive inference -- we plot their attention masks. (since NeurIPS does not allow authors to upload an image or provide an external link, we put the python code here and it'll plot the masks)
>
> ```python
> import torch
> import matplotlib.pyplot as plt
>
> patch_nums = [1, 2, 3, 4, 5, 6, 8, 10, 13, 16]
> L = sum(pn**2 for pn in patch_nums)
> d = torch.cat([torch.full((pn * pn,), i) for i, pn in enumerate(patch_nums)]).view(1, L, 1)
> dT = d.transpose(1, 2)
> mask_VAR = (d >= dT).reshape(L, L).numpy()
>
> mask_AR = torch.tril(torch.ones(patch_nums[-1]**2, patch_nums[-1]**2)).numpy()
>
> fig, axs = plt.subplots(1, 2, figsize=(10, 5))
> axs[0].imshow(mask_VAR), axs[0].set_title('mask_VAR'), axs[0].axis('off')
> axs[1].imshow(mask_AR), axs[1].set_title('mask_AR'), axs[1].axis('off')
>
> plt.show()
> ```
>
> From the figure one can see that:
>
> - VAR's block-wise causal mask and VQGAN's standard causal mask look very close.
> - Or in other words, VAR still maintains many AR properties.
> - So both VQGAN and VAR can benifit from the batched inference, and their batch size sweet-spots (when the throughput starts saturating) can be close to each other.
>
> &nbsp;
>
> We will add all of the above supplementary results and analysis to a new Appendix section named throughput benchmark using batched inference. Thank you again for your constructive and detailed response.

---

> > ### Comment · Reviewer_oCJa · 2024-08-12
> >
> > Thanks for engaging throughout the rebuttal period.
> > I am now happy with any issues I originally had with the paper and will keep my score as is.
> >
> > I would encourage the authors to perform some additional analysis on the throughput evaluations if they have time, however, I won't ask for more during this rebuttal period as it feels like chasing small details. I am still a little confused as to how VAR has better throughput compared to VQGAN given that VQGAN has significantly fewer tokens and lower attention cost according to the mask presented (half of the last scale of VAR).

---

### Official Review · Reviewer_Hisd · 2024-07-12

**Soundness:** 3
**Presentation:** 3
**Contribution:** 4
**Rating:** 7
**Confidence:** 4

**Summary:**

This paper introduces next-scale prediction autoregressive models that satisfy mathematical premises (unidirectional dependency of autoregressive model) and preserve the 2D spatial locality. The core method is to develop multiscale VQ-VAE. The proposed method is more efficient than the traditional autoregressive model, requiring only $O(n^4)$ compared to $O(n^6)$ of the raster autoregressive model. Furthermore, this method is proven to follow scaling laws of LLM which guarantee better performance when scaling up the training process.

**Strengths:**

1. This idea is natural and novel. It successfully solves mathematical premises violation of previous rastering scan autoregressive model.
2. The power-law scaling law is interesting and encourages follow-up work to scale up models for better performance.
3. The method’s performance is competitive to the diffusion model and other generative models.
4. Most of the paper claims seem valid to me.

**Weaknesses:**

1. The second stage of training VAR transformers is too short and is hard for me to fully understand how it works. I wonder about the details of how to generate $h_w \times w_h$ tokens in $r_k$ parallel using k-th position embedding map. Is the embedding 1D or 2D embedding ?. How to make sure all tokens in $r_k$ are correlated to each other ?
2. The highest resolution of the scale $r_K$ is $16 \times 16$ and there are 10 scales (1,2,3,4,5,6,8,10,13,16). I wonder if there is any motivation to choose these scales.
3. I think the paper should include the sampling algorithm of autoregressive model with hyper-parameter details such as temperature, top-k, top-p and CFG sampling.
4. The zero-shot generalisation algorithm should be included in the paper for clarity.

**Questions:**

My main concerns are in the method section. I hope the author could provide more method details. See the weakness above.

**Limitations:**

The limitation discussions are sound and clear to me.

---

> ### Author Rebuttal · Authors · 2024-08-07
>
> Dear Reviewer Hisd,
>
> Many thanks to your valuable comments and questions, which help us a lot to improve our work. We address your questions as follows.
>
> ---------------------------
>
> > [W1] The second stage of training VAR transformers is too short and is hard for me to fully understand how it works...
>
> [A2] Thanks for pointing this and we'll detail more on the 2nd stage of VAR method in our manuscript. To generate $h_k\times w_k$ tokens $r_k$ in $k$-th autoregressive step (like to generate $r_2$ in Figure 4), the input is the previous token map $r_1$. It'll be reshaped to 2D ($1\times 1$ here), embedded to a 2D feature map, upsampled to $r_2$'s shape $2 \times 2$, projected to the VAR transformer's hidden dimension, and added by a 2D positional embedding to get $e_1$. So the actual input to the VAR transformer is $e_1$ in Figure 4, and these steps are the details of "word embedding and up-interpolation" noted in Figure 4. The detailed implementation can also be found in the code "codes/models/var.py autoregressive_infer_cfg" attached in supplementary material.
>
> To make sure all tokens in $r_k$ are correlated to each other, we don't apply attention mask within $r_k$ to keep a bidirectional attention on them. In other words, $r_k$ can attend to all tokens of $r_{\le k}$
>
> &nbsp;
>
> > [W2] there are 10 scales (1,2,3,4,5,6,8,10,13,16). I wonder if there is any motivation to choose these scales.
>
> [A2] Yes the chose of scales is a key design of VAR. We choose an *exponential function* $h_k=w_k=\lfloor a\cdot b^k\rfloor$ to get these scales because:
> 1) As discussed in appendix F, we can reach a total complexity of $\mathcal{O}(n^4)$ via an exponential schedule.
> 2) We want to increase the number of steps to reach $16\times 16$ for image quality. An exponential function grows slowly in the early stages and quickly in the later ones. So it allows us to increase steps (mainly in the early stages) without a significant increase in total sequence length.
>
> In practice, we use $a=1.36$ and $b=1.28$ to get that (1,2,3,4,5,6,8,10,13,16).
>
> &nbsp;
>
> > [W3] I think the paper should include the sampling algorithm of autoregressive model with hyper-parameter details such as temperature, top-k, top-p and CFG sampling.
>
> [A3] We use temperature of $1.0$, top-k of $k=900$, top-p of $p=0.96$, and CFG of $1.25$ (VAR-$d16$) or $1.5$ (the others). We'll add these to our manuscript.
>
> &nbsp;
>
> > [W4] The zero-shot generalisation algorithm should be included in the paper for clarity.
>
> [A4] Thanks for this valuable advice. We'll add a pseudo code for detailing the algorithm of zero-shot generalisation in our manuscript.
> The algorithms of inpainting, outpainting, and class-condition editing are basically the same.
>
> Specifically, we would mask out the according region at each scale of (1,2,3,4,5,6,8,10,13,16) given the task, i.e., masking the inner area for inpainting, the outer area for outpainting, and the area we want to edit for editing. During VAR generation, ground-truth non-masked tokens are maintained (like teacher forcing) and we only collect VAR's predictions on those masked regions.
>
> -----------------------------------
>
> **Thank you again for helping us improve the paper and hope our response can resolve your concerns!
> Please let us know if you have any further questions.
> We will be actively available until the end of rebuttal period.
> If you feel your concerns are addressed, please consider reevaluating our work.
> Looking forward to hearing from you :-) !**

---

> > ### Comment · Reviewer_Hisd · 2024-08-08
> >
> > Thanks for addressing my concerns. I decided to raise score to 7

---

> ### Author Response · Authors · 2024-08-09
> **We appreciate your acknowledgement on our rebuttal and the reevaluation**
>
> We're glad to hear your concerns have been addressed. We'll be active till the end of the discussion period. If you have more questions, please let us know. Thank you!

---

### Author Response · Authors · 2024-08-07
**Overall Author Rebuttal**

We sincerely thank every reviewers and chairs for their time and effort.
We appreciate it a lot that our **novelty, motivation, decent performance, scaling law properties, and a lot of technical contributions** have been acknowledged by the reviewers.

Here, we respond to two common questions about VQVAE and zero-shot algorithm respectively.

> [C1] VQVAE concerns

1. how to choose the 2D shape of the latent token map at each scale: we choose (1,2,3,4,5,6,8,10,13,16) for 256px image and (1,2,3,4,6,9,13,18,24,32) for 512px image. the chose of scales is a key design of VAR. We choose an *exponential function* $h_k=w_k=\lfloor a\cdot b^k\rfloor$ to get these scales because:
    - As discussed in Appendix F, we can reach a total complexity of $\mathcal{O}(n^4)$ via an exponential schedule.
    - We want to increase the number of steps to reach $16\times 16$ for image quality. An exponential function grows slowly in the early stages and quickly in the later ones. So it allows us to increase steps (mainly in the early stages) without a significant increase in total sequence length.
    - In practice, we use $a=1.36$ and $b=1.28$ to get that (1,2,3,4,5,6,8,10,13,16).
    - We've added these explanations to our manuscript.
2. reconstruction evaluation: We added an evaluation of rFID to a new Appendix section of "Comparisons on VAEs" and also pasted it here. It evaluated different VAEs on the 256px ImageNet validation set. Generally, the number of tokens is crucial to a VAE's performance, which shows a trade-off between sequence length and reconstruction FID (rFID). VAR-VQVAE with 680 tokens performs similarly to VQGAN's with 1024 tokens.
| Model | downsampling| num of tokens | rFID$\downarrow$|
|:-|:-|:-|:-|
| VQGAN-VQVAE | 16 | 256   | 4.90 |
| VQGAN-VQVAE | 8  | 1024  | 1.14 |
| MaskGIT-VQVAE  | 16 | 256  | 2.28 |
| VAR-VQVAE   | 16 | 680   | 1.00 |
| SDXL-VAE    | 8  | 1024  | **0.68** |
3. how to choose the number of scales: see above 1.
4. code usage of each scale: we counted the code usage of our VQVAE on the 256px ImageNet validation set separately for each scale, and found each scale had a code usage >99%. We've added this to our paper.
5. do different scales capture similar (spatial) information in the latent space vs the image space: we observed that small scales seemed to encode image's low-frequency components while large scales represented high-frequency details. We have added a new Appendix section "Per-scale visualization of VAR VQVAE" to show this figure.
6. VQVAE ablation: Since this work mainly aims to explore a new VAR algorithm, we keep the VQVAE structure as simple as possible. Specifically in line173, we use the same architecture of VQGAN's single-scale VQVAE and the multi-scale modules (several convolutions) only contain 0.03M parameters. So comparing VQGAN-VQVAE and our VQVAE can be seen as an ablation of this VQVAE. We've added these to our paper.
7. details to reproduce the VQVAE: we've added more hyperparameters and training costs in Appendix B: Implementation details, which are: 16 epochs training on OpenImages, batch size 768, fp16 precision by `torch.cuda.amp`, standard AdamW optimizer with betas (0.5, 0.9), learning rate 2e-4 and weight decay 0.005. The loss weight of L2 reconstruction, L1 reconstruction, codebook loss, commitment loss, LPIPS and discriminator are 1.0, 0.2, 1.0, 0.25, 1.0, 0.4. This training will take ~60h on 16 A100 80G GPUs.


&nbsp;

> [C2] Zero-shot generalisation algorithm

We'll add a pseudo code for detailing the algorithm of zero-shot generalisation in our manuscript.
The algorithms of inpainting, outpainting, and class-condition editing are basically the same.

Specifically, we would mask out the according region at each scale of (1,2,3,4,5,6,8,10,13,16) given the task, i.e., masking the inner area for inpainting, the outer area for outpainting, and the area we want to edit for editing.
By "according", we mean some interpolation should be used. E.g., if we want to mask the upper-left 128x128 area of a 256x256 image then do inpainting, the upper-left area of **each scale token map** would be masked.
During each VAR step, non-masked tokens are teacher forced, and VAR only needs to predict tokens on those masked regions.

-----------------------
For non-common questions, we've responded below the review comment of each reviewer.


Best,

The Authors

---

> ### Comment · Reviewer_yx9H · 2024-08-08
> **Revised Manuscript?**
>
> It seems that the manuscript does not contain your added information. Does NeurIPS allow revising manuscript during rebuttal period?

---

> ### Author Response · Authors · 2024-08-08
> **Sorry NeurIPS2024 does not allow paper revision; We provide more detailed evidence and responses for you**
>
> Dear Reviewer yx9H,
>
> 1. In the "Author responses" part at https://neurips.cc/Conferences/2024/CallForPapers , it is said that "Authors may not submit revisions of their paper or supplemental material, but may post their responses as a discussion in OpenReview."
> This policy does cause some inconvenience, but we have done our best to **describe all revisions and copied all new experimental results** to our responses.
>
> &nbsp;
>
> 2. We went through each of your questions and our responses again, and found that **only one revision** ([W2] Resolution Flexibility) was not copied to our response. So we provide more evidence for you:
>
>     - The VAR transformers generating 256 and 512 images both use the same multi-scale VQVAE that is trained only on 256 resolution. The good performance of the VAR on the 512 generation task shows the generalizability of VQVAE.
>
>     - This 256-trained multi-scale VQVAE has a good reconstruction FID of 2.28 on the 512 ImageNet validation set.
>
> &nbsp;
>
> 3. For your convenience, here we also provide a complete response to your [Q1] question about VQVAE, so you don't need to search them in the Overall Author Rebuttal:
>
> - VQVAE rFID: We added an evaluation of rFID to a new Appendix section of "Comparisons on VAEs" and also pasted it here. It evaluated different VAEs on the 256px ImageNet validation set. Generally, the number of tokens is crucial to a VAE's performance, which shows a trade-off between sequence length and reconstruction FID (rFID). VAR-VQVAE with 680 tokens performs similarly to VQGAN's with 1024 tokens.
> | Model | downsampling| num of tokens | rFID$\downarrow$|
> |:-|:-|:-|:-|
> | VQGAN-VQVAE | 16 | 256   | 4.90 |
> | VQGAN-VQVAE | 8  | 1024  | 1.14 |
> | MaskGIT-VQVAE  | 16 | 256  | 2.28 |
> | VAR-VQVAE   | 16 | 680   | 1.00 |
> | SDXL-VAE    | 8  | 1024  | **0.68** |
>
> - pre-training cost: we've added more hyperparameters and training costs in Appendix B: Implementation details, which are: 16 epochs training on OpenImages, batch size 768, fp16 precision by `torch.cuda.amp`, standard AdamW optimizer with betas (0.5, 0.9), learning rate 2e-4 and weight decay 0.005. The loss weight of L2 reconstruction, L1 reconstruction, codebook loss, commitment loss, LPIPS and discriminator are 1.0, 0.2, 1.0, 0.25, 1.0, 0.4. This training will take ~60h on 16 A100 80G GPUs.
>
> - why we didn't choose other multi-scale quantization ways: if the residual quantization is removed and independently quantization is used (e.g., independently downsampling VAE features, or independently encoding images in 16x16, 32x32, ..., 256x256), the unidirectional dependency of VAR algorithm would be broken. The details can be found in line140 of our paper. To ensure that property, we have to use the residual quantization. We'll add these explanations to our manuscript.
>
> - Lack of Ablation Study on VQVAE: Since this work mainly aims to explore a new VAR algorithm, we keep the VQVAE structure as simple as possible. Specifically in line173, we use the same architecture of VQGAN's single-scale VQVAE and the multi-scale modules (several convolutions) only contain 0.03M parameters. So comparing VQGAN-VQVAE and our VQVAE can be seen as an ablation of this VQVAE. We've added these to our paper.
>
>
> &nbsp;
>
> We hope the above response can help solve your questions.
> Thanks again for your thorough review and looking forward to your reply!

---

### Public Comment · ~Philip_Bachman1 · 2024-11-21
**Autoregression and Bayes' rule**

Thanks for the nice paper. Your statements on page 4 concerning a "unidirectional dependency assumption" in autoregressive models are incorrect. Fixed order autoregressive models assume some order in which to decompose a joint distribution into a sequence of simpler conditional distributions following Bayes' rule. If we assume the correctness of Bayes' rule, which seems reasonable, then the product of the simpler conditionals is precisely equivalent to the full joint distribution. Ie, autoregression does not require any particular dependency structure among the variables in the joint distribution for correctness.

I suspect that you're slightly misstating the intuition that some sequential decompositions might be easier to learn or otherwise work with than others. Eg, in systems with strong "causal" structure it might make sense to predict effects conditioned on causes rather than predicting causes conditioned on effects. There are scenarios in which this intuition is definitely correct, ie, choosing the "correct" ordering for the autoregression offers formal benefits over the "wrong" ordering. However, the "wrong" ordering is still capable, in principle, of representing the full joint distribution faithfully.

---

### Decision · Program_Chairs · 2024-09-25

**Decision:**

Accept (oral)

**Comment:**

This paper introduces a novel visual autoregressive model (VAR) that predicts the next image scale rather than rasterizing the image pixel-wise. The VAR model shows strong results in image generation, outperforming existing autoregressive models in efficiency and achieving competitive results with diffusion-based methods.

Strengths:
* Clear Motivation: The motivation behind multiscale image generation is well-articulated, making it a natural and novel extension of existing autoregressive techniques.
* Novelty & Methodology: The paper presents a fresh approach to autoregressive image generation by adopting a coarse-to-fine multiscale framework, which offers new insights into how autoregressive models can be applied in vision tasks.
* Strong Empirical Results: The model demonstrates strong performance on ImageNet in terms of generation quality, diversity, and inference speed, with impressive FID and IS scores. The scaling law properties are promising, showing room for future advancements.
* Efficiency: VAR achieves a significant improvement in inference speed, leveraging its multiscale approach to reduce computational costs while maintaining high-quality image generation.

Weaknesses:
* Missing Ablation Studies: Several reviewers expressed concerns over the lack of ablation studies on the VQ-VAE component, specifically regarding residual quantization and its impact on VAR's performance.
* Clarification of Details: The method section, particularly concerning VAR’s training and the residual tokenization process, could be better clarified. Some key details about the VQVAE's role and scale selection could benefit from deeper discussion in the main paper rather than being relegated to appendices.
* Comparison to Stronger Baselines: Although VAR performs well against the baselines provided, the baselines for diffusion models could have been stronger. Comparisons to models like MDTv2 were suggested to be more appropriate.
* The authors have not verified the scalability of the approach beyond ImageNet, where the performance of AR vs VAR can shift as we scale up the data and model.

The authors provided a comprehensive rebuttal, addressing key concerns raised by the reviewers. They acknowledged the lack of ablation on residual quantization and provided an analysis on why this component is crucial to VAR’s success. Additionally, the authors clarified technical details related to scale selection, VAR’s inference steps, and the interaction between scales in image editing tasks. The revised manuscript will include more details on these points, as well as additional comparisons with state-of-the-art diffusion models. Despite minor weaknesses, the contributions and technical soundness of VAR make it a valuable addition to the field of autoregressive models for image generation. Based on above, I recommend accepting this paper.